# Reversible and controllable reduction in friction of atomically thin two-dimensional materials through high-stress pre-rubbing

Haoyang Su [1,3], Honglin Zhang[2,3], Junhui Sun [2,3], Haojie Lang [1], Kun Zou[1] & Yitian Peng [1]✉

Great efforts have been made to further reduce friction of atomically thin two-dimensional (2D) materials as solid lubricants due to their exceptional tribological properties and mechanical strength. In this work, the friction of atomically thin graphene is extensively and controllably reduced through pre-rubbing under high stress, resulting in a reduction of the friction coefficient by up to a factor of six compared to the pristine graphene. Also, this reduction can be reversed by reciprocating friction under moderate stress. Furthermore, high-stress pre-rubbing allows for patterning intentionally lubricating features on atomically thin graphene, such as nanometer-sized letters. This reduction in friction is attributed to the decreased sliding potential barrier yet increased contact stiffness, induced by the enhanced strength of graphene adhesion to the substrate due to interfacial charge transfer, as revealed by density functional theory (DFT) calculations. These findings present a practical methodology for optimizing and controlling the performance of 2D materials.

Friction is an inherent challenge in virtually all mechanical systems with moving components, causing undesired energy dissipation and component failures. It has been estimated that one-third to one-half of the world's energy is dissipated through various forms of friction[1]. Therefore, it is critical to develop strategies for restraining friction. The emergence of atomically thin 2D materials has sparked immense interest due to their exceptional mechanical and tribological properties. Graphene, for instance, demonstrates unprecedented mechanical strength with Young's modulus of 1 TPa[2], tensile strength of approximately 130 GPa[3], and exceptional lubricity characterized by a typical friction coefficient of 0.01[4]. These distinctive structural and outstanding tribological characteristics of atomically thin 2D materials render them widely used as solid lubricants and protective films, improving the reliability and lifespan of sliding components.

Despite the exceptional frictional properties of atomically thin 2D materials, their frictional performance is influenced by multiple factors, such as thickness, crystalline orientation, environmental conditions, substrate properties, etc[5]. Specifically, combined with the inherently high out-of-plane flexibility of atomically thin 2D materials, the weak interaction between 2D materials and the substrate, mediated by van der Waals forces, significantly increases their surface friction[6]. Besides, the surface friction behaviors of atomically thin 2D materials depend on the competition between tip-to-flake and flake-to-substrate interactions, with friction decreasing as the flake-to-substrate interaction increases[7]. Consequently, it remains to be a great challenge to reduce friction on atomically thin 2D materials further. Several researchers have explored various approaches to address this issue. For instance, using nanotextured substrates, ultralow friction in graphene is achieved through strain engineering[8]. Conducting different durations of plasma treatment to the $SiO_2/Si$ substrate achieved varying degrees of friction reduction, attributed to enhanced interfacial adhesion[9]. Constructing sliding interfaces with 2D materials can even achieve a superlubric state, owing to the exceptionally low shear strength under incommensurable contact[10]. Although existing studies have provided some

[1]College of Mechanical Engineering, Donghua University, Shanghai 201620, China. [2]Tribology Research Institute, State Key Laboratory of Traction Power, School of Mechanical Engineering, Southwest Jiaotong University, Chengdu 610031, China. [3]These authors contributed equally: Haoyang Su, Honglin Zhang, Junhui Sun. ✉e-mail: yitianpeng@dhu.edu.cn

approaches to reduce the friction of atomically thin 2D materials, most of these approaches are either difficult to apply in practice or uncontrollable.

In this work, a practical approach has been proposed to extensively and controllably reduce the friction of graphene through high-stress pre-rubbing. Impressively, the observed friction reduction can be reversed by reciprocating rubbing under moderate stress. Additionally, intentional patterns of nanoscale lubrication features can be precisely achieved on graphene surfaces. DFT calculations demonstrate the pre-rubbing induced friction decline may arise from the reduction in sliding potential barrier and the increase in contact stiffness, both of which are induced by enhanced interfacial adhesion strength due to charge transfer during high-stress pre-rubbing.

## Results

### High-stress pre-rubbing and friction measurements

As depicted in Fig. 1a, the pre-rubbing process involves rubbing graphene flakes deposited on a $SiO_2/Si$ substrate using a diamond tip at a given stress. Figure 1b displays the zoomed-out topography encompassing the areas pre-rubbed under 11.6 GPa. High-stress pre-rubbing results in the sinking of the surface, forming a crater. Figure 1c presents the friction mapping for the tip sliding on the graphene, revealing a noticeable reduction within the pre-rubbed region compared to the pristine area. Further experiments conducted on $SiO_2/Si$ supported $MoS_2$ were shown in Supplementary Fig. 1, with similar reductions in friction suggesting that this approach might be a universal strategy for 2D materials.

Then, pre-rubbing under various stress levels was conducted to evaluate the impact of different stresses on topography and tribological characteristics. It should be noted that at stress levels of 7.8 GPa and below, no visible change in morphology and friction is observed, the sinking height increases with the pre-rubbing stress, progressively reaching 2.94 nm after pre-rubbing under 13.2 GPa, as shown in Supplementary Fig. 2. The initial thickness of the graphene flakes before pre-rubbing was measured at approximately 0.98 nm and confirmed to be a monolayer structure by Raman spectroscopy. In general, the thickness of monolayer graphene deposited on a $SiO_2$ substrate was measured to be in the range of 0.6 to 1.2 nm due to variations in interfacial interactions between the substrate and graphene[11]. This magnitude change in sinking height indicates that deformation occurs in the pre-rubbed region.

Atomic-scale friction behavior in both the pristine region and the pre-rubbed region was examined using a Cypher ES AFM instrument. As shown in Fig. 1d, the irregular lattice together with intrinsic ripples can be seen from atomic-scale lateral force mapping in the pristine region, which may result from the surface fluctuations of graphene loosely adhered to the $SiO_2/Si$ substrate. In the stick-slip friction behavior of the pristine graphene, as shown in Fig. 1e, the friction force increases with distance during the initial sliding, reaching saturation after approximately 3 nm of sliding, exhibiting a pronounced strengthening effect (slope). Conversely, as shown in Fig. 1f, the atomic-scale lateral forces remarkably decrease for the graphene region after pre-rubbing under 11.6 GPa stress (lattice information acquired using a fast Fourier transform, shown in the inset). The six-membered ring structure can be distinguished from the lateral force maps on the pre-rubbed area, the graphene lattice is distinctly visible, and the ripples are significantly suppressed. Notably, as shown in Fig. 1g, the strengthening effect in the stick-slip friction is substantially reduced.

For the characterization of variations in tribological properties, friction tests were conducted in distinct pre-rubbed areas. A 500 nm long trace is selected in the pre-rubbed area and repeatedly scanned to conduct a friction test, while gradually reducing the load until the probe is slid off the sample. The load change ranged from 33 nN to −10

nN (corresponding maximum stress 1.4 GPa), and three tests were conducted at 100 nm intervals in each pre-rubbed area. Figure 1h illustrates the load-dependent friction in the pre-rubbed area under different stress levels. The topography and localized friction mapping after the pre-rubbing process was presented in Supplementary Fig. 2 to show the comparison of sinking height and friction after pre-rubbing under different stresses. The fitted coefficient of friction is depicted in Fig. 1i. Generally, the coefficient of friction gradually decreases with the increase of pre-rubbing stress. Notably, after pre-rubbing under 13.2 GPa, the friction coefficient is reduced to 0.0022, which is one-sixth of the value on the pristine graphene surface. Ultra-low friction state (friction coefficient below 0.01)[12] are achieved through pre-rubbing under 11.6 GPa or higher stresses.

### Friction measurements of different thicknesses after pre-rubbing

Pre-rubbing experiments were further conducted on graphene with varying thicknesses under a stress level of 11.6 GPa. Figure 2a presents the friction mappings of graphene with different thicknesses after pre-rubbing. The topography and cross-sectional data profiles of Fig. 2a are shown in Supplementary Fig. 3. The labels in Fig. 2a correspond to the layer numbers of graphene, for example, the region labeled as '1 L' corresponds to monolayer graphene. The layer number of multilayer graphene is determined by dividing the height difference by the theoretical thickness of a monolayer of graphene ($0.35 \pm 0.01 \, nm$[13]), combined with the Raman spectra and the Lorentz multi-peak fitting[14] shown in Fig. 2b and Fig. 2c. The cross-sectional data along the red dashed line in Fig. 2a and the average friction of both pristine graphene and graphene pre-rubbed under 11.6 GPa with distinct thicknesses are illustrated in Fig. 2d, e. In the pristine regions, compared to 1-layer graphene, the friction is ~13% lower on 2-layer graphene and 23% lower for 3-layer graphene. This indicates that friction decreases with increasing thickness, showcasing an obvious thickness dependence. Notably, the friction differences among 1-layer, 2-layer, and 3-layer graphene after pre-rubbing are within 5%, indicating that the friction in the pre-rubbed region has less dependence on thickness compared with the pristine region.

### Friction Recovery in the pre-rubbed region

What is particularly interesting is that this reduction in friction can be reversible through reciprocating friction under moderate stress. Figure 3a presents friction force mappings obtained from reciprocating friction at 5.6 GPa, within an expanded region that includes the area previously subjected to high-stress pre-rubbing at 11.6 GPa. The results indicate that during the reciprocating friction process, the friction force at the edge of the pre-rubbed region gradually restores consistency with the surrounding area, with the restored region expanding as reciprocating friction continues. After 35 cycles of sliding, the friction force in the pre-rubbed region is fully restored and slightly exceeds that of the original graphene region. Figure 3b and Fig. 3c display the line profile data of the friction force and the corresponding height, respectively. As reciprocating friction progresses, the friction force at the edge of the pre-rubbed region transitions from low to match the surrounding area, with corresponding changes in the sinking height. For instance, at the 9th and 21st cycles, changes in both friction force and sinking high are observed on the left side of the pre-rubbed region, as indicated by the red dashed line and blue dashed line. After 35 cycles of sliding, the friction force in the pre-rubbed region is fully restored, and the corresponding sinking height decreases by 0.15 nm.

## Discussion

The Scanning Kelvin Probe Microscopy (SKPM) was conducted to analyze the changes at the interface. Figure 4a shows the SKPM images of the graphene regions pre-rubbed under the stress below at 13.9 GPa

and 14.6 GPa for friction modulation, and Fig. 4b presents the surface work function. It is evident that the work function increases after pre-rubbing at stress levels of 9.2 GPa and above.

The variation in contact potential difference can be converted into a change in the Fermi level using the following expression[15]:

$$\Delta E_F = e\Delta V_{CPD} \tag{1}$$

where $e$ is the elementary charge, and $\Delta V_{CPD}$ is the change in CPD relative to pristine graphene. Meanwhile, the change in carrier concentration in the modified graphene region can be estimated as[16]:

$$\Delta n = \frac{1}{\pi}\left(\frac{\Delta E_F}{h v_F}\right)^2 \tag{2}$$

Here the $h$ is the Planck constant, and $v_F$ is the Fermi velocity. The calculated changes in the Fermi level and carrier concentration are shown in Fig. 4c. It can be observed that the Fermi level of graphene gradually shifts downward with increasing pre-rubbing stress, and the carrier concentration gradually increases. The results of DFT calculations also demonstrate that pre-rubbing leads to a decrease in the Fermi level of graphene/substrate, as shown in Supplementary Fig. 4.

We perform DFT calculations to investigate the interfacial charge distribution, adhesion strength of graphene on substrate, stiffness, and the sliding energy barrier for tip on the graphene-substrate towards the pre-rubbed induced friction reduction. The results shown in Fig. 4e indicate the charge transfer between graphene and the substrate increase with indentation (compress) distance. The results are consistent with the results of SKPM measurements shown in

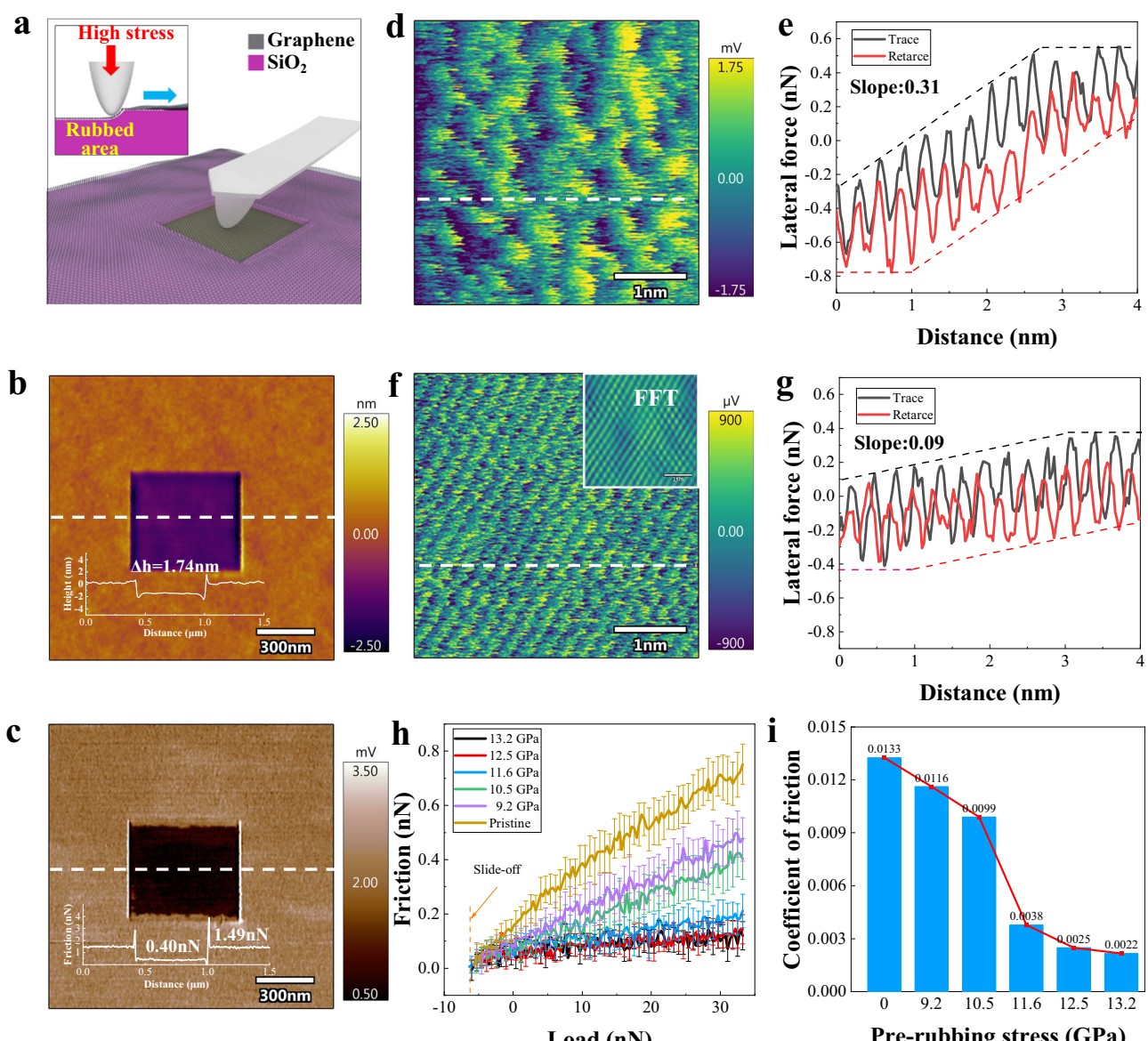

**Fig. 1 | High-stress pre-rubbing induced ultra-low friction of graphene.**
**a** Schematic diagram of the pre-rubbing process under high stress and the friction measurement. **b** Topography image and (**c**) friction mapping after pre-rubbing of 11.6 GPa. The insets in (**b**) and (**c**) show the topography data and friction force data along the white dashed line, respectively. $\triangle$h indicates the height difference. **d** High-resolution lateral force mapping for pristine graphene and (**f**) for pre-rubbed graphene. The inset in (**f**) shows the output results of the fast Fourier transform (FFT) applied to the pre-rubbed region. **e, g** Representative variation of the friction stick-slip along the white dashed lines shown in (**d**) and (**f**), respectively, the slope reflects the degree of friction enhancement. **h** Friction as a function of normal load under different pre-rubbing stress, with error bars indicating the standard deviation of the data based on 256 data points. **i** Pre-rubbing modulated coefficient of friction fitted from (**h**). Source data of (**b, c, e, g–i**) are provided as a Source Data file.

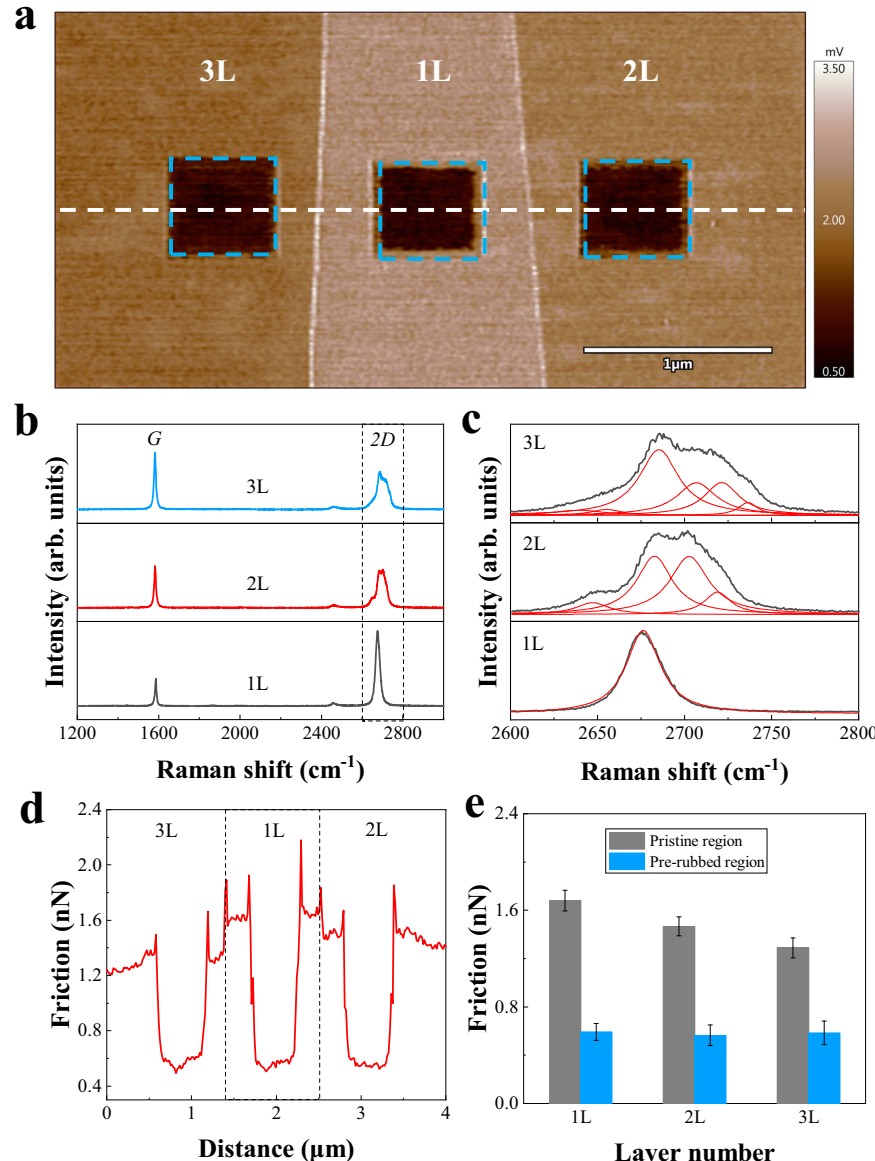

**Fig. 2 | Thickness dependence of friction on graphene after pre-rubbing.**
**a** Friction force mapping of the distinct thickness of graphene after pre-rubbing with identical parameters, the blue dashed box indicates the pre-rubbed region, and the inset labels correspond to the layer numbers of graphene. **b** Measured Raman spectra from different regions. **c** Corresponding fitted Lorentzian curves of the 2D feature from (**b**). **d** Cross-sectional data profiles along the white dashed lines in (**a**). **e** Average data of the friction in distinct areas, with error bars representing the standard deviation based on 900 data points. Source data of (**b**–**e**) are provided as a Source Data file.

Fig. 4c, because the charge transfer at the graphene-substrate interface generally causes doping of the graphene layer[17]. This would lead to an increased adhesion of graphene on the substrate (Fig. 4e, right side)[18]. The interaction strength between the graphene and substrate atoms is also discussed by the crystal orbital Hamiltonian population (COHP) shown in Supplementary Fig. 5. It also implied that pre-rubbing leads to increased adhesion of graphene on the substrate. Further, as shown in Fig. 4f, the stiffness of tip/graphene-substrate increases due to enhanced adhesion of graphene/substrate. The reactivity of graphene decreases due to its decreased deformation[19]. Meanwhile, the sliding potential corrugations (Fig. 4f, right side) for tip on graphene also decrease due to the increased interaction between the graphene and substrate[20]. According to the PT model[21], the increased stiffness and decreased sliding potential corrugations would restrain friction of the tip sliding on the graphene-substrate.

Additionally, this hypothesis above is also supported by thickness-dependent experiments of atomic stick-slip friction behavior (Fig. 2) as well as the friction revivification experiments (Fig. 3). The out-of-plane stiffness of graphene increasing with thickness may suppress the puckering effect and result in a reduction in the contact area during sliding[22]. Friction revivification experiments further indicate that graphene is tightly adhered to the substrate following high-stress pre-rubbing. The increased graphene-substrate interfacial adhesion strength induces a significant reduction in surface friction in the pre-rubbed region. However, after reciprocating rubbing under moderate stress over an enlarged area containing the high-stress pre-rubbed region, the interface distance between the graphene and substrate increases (inferred from the reduction in sinking height, Fig. 3c. This suggests a reduction in adhesion strength between the graphene and the substrate, resulting in the inability of the pre-rubbed region to

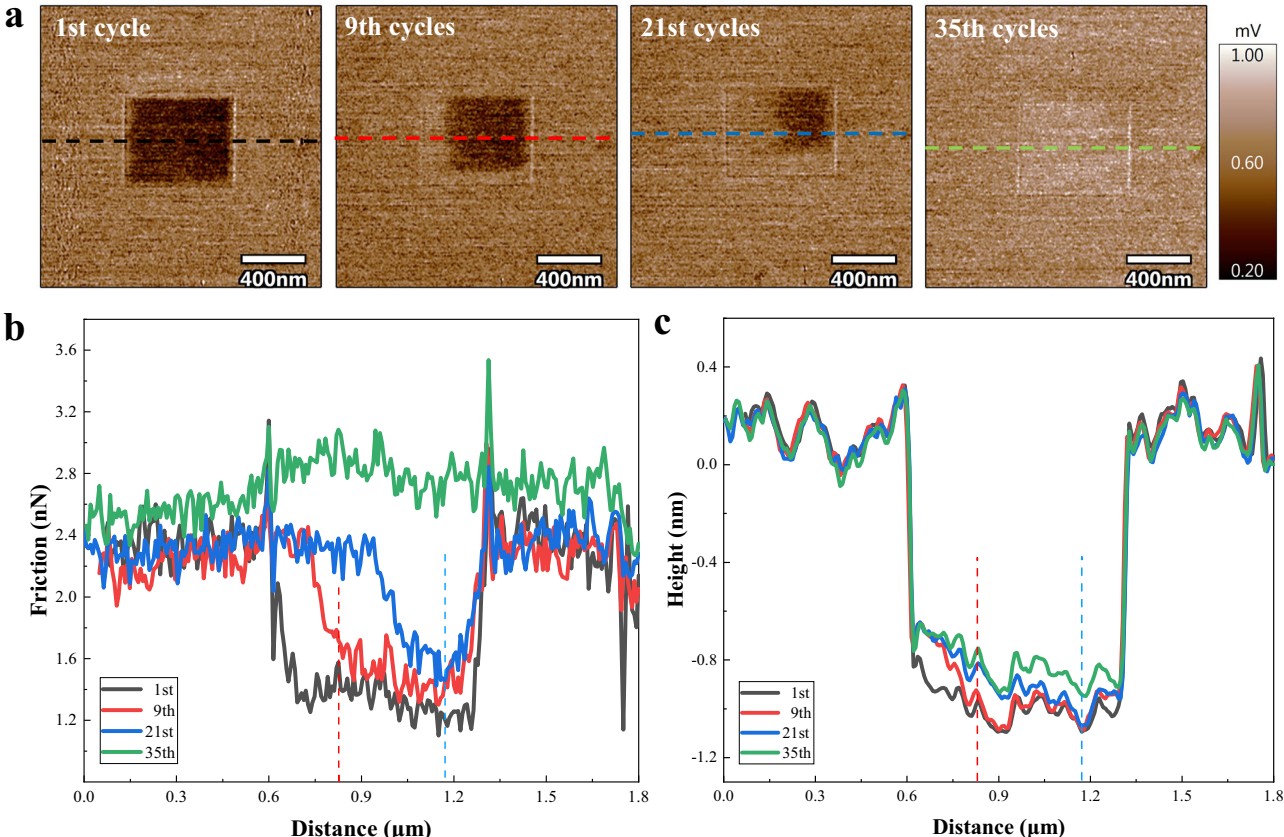

**Fig. 3 | Friction revivification experiments. a** Friction mapping of the reciprocating scan cycles of 1 time, 9, 21, and 35 times. **b, c** Shows the friction data profiles and height data profiles along the dashed line in (**a**). The color of the friction data corresponds to that in (**a**). Source data of (**b**), and (**c**) are provided as a Source Data file.

maintain the reduced friction level, causing the friction force to revert to that of pristine graphene.

With nanometer-scale precision, the high-stress pre-rubbing process has been employed to create intricate patterns on graphene surfaces. This technique enables the selective generation of lubricating features, including squares with varying side lengths ranging from 200 to 600 nanometers, as illustrated in Fig. 5a. The minimum pattern width can be tailored to be as small as the tip radius, as depicted in Fig. 5b. Notably, intentional patterning of nano scale specific letters, such as "2D" for 2D materials and "DHU" for Donghua University, has been successfully demonstrated on the graphene surface, as shown in Fig. 5c, d. Considering that AFMs can be fully integrated on a single chip[23], this ability to design and implement nanoscale friction patterns and revivification holds transformative potential for a variety of cutting-edge technologies. For instance, on the one hand, the precise control over friction at the nanoscale holds substantial promise for optimizing the performance and durability of micro- and nano-electromechanical systems (MEMS/NEMS)[24], where friction management is critical. On the other hand, encoding and erasing frictional states in designated patterns paves the way for the development of advanced information storage systems based on the tribological properties of atomically thin 2D materials[6].

In summary, the friction of atomically thin 2D materials has been significantly reduced through pre-rubbing under high stress. The friction properties can be controlled by pre-rubbing under varying stress levels, the friction coefficient of graphene is reduced by a factor of six at the greatest extent. This reduction in friction can be reversible through reciprocating friction under moderate stress. Additionally, the high-stress pre-rubbing process was employed to pattern on the

graphene surfaces intentionally, creating nanoscale lubricating features. DFT calculations reveal that the friction reduction is due to a combination of a decreased sliding potential barrier and increased contact stiffness, both induced by enhanced graphene-substrate adhesion strength due to interfacial charge transfer during high-stress pre-rubbing. This research provides a practical methodology for optimizing and controlling the performance of atomically thin 2D materials, with significant implications for applications in MEMS/NEMS and nano-fabrication.

## Methods
### Fabrication of graphene
Graphene flakes were mechanically exfoliated from commercial natural graphite using the scotch-tape method and subsequently deposited on heavily p-type doped silicon substrates with a 300 nm thick silicon dioxide ($SiO_2$/Si) layer. Few-layer graphene regions were identified by distinct colors under an optical microscope, and subsequently, the thickness of the graphene flakes was measured in noncontact mode AFM (MFP-3D, Asylum Research) to select the test area.

### Pre-rubbing process and friction
The pre-rubbing process was achieved by rubbing graphene flakes under desired stress using the AFM instrument with a diamond tip (FM-LC, 10 N/m, Adama Innovations) in contact mode, consisting of the following steps: first, a gentle load of 20 nN was exerted to the tip, scanning over a large area to ensure the absence of contaminants on the graphene surface; second, the tip was brought into contact with graphene at a load corresponding to the given stress; finally, the tip scanned a square area with a side length of 600 nm twice at a scan

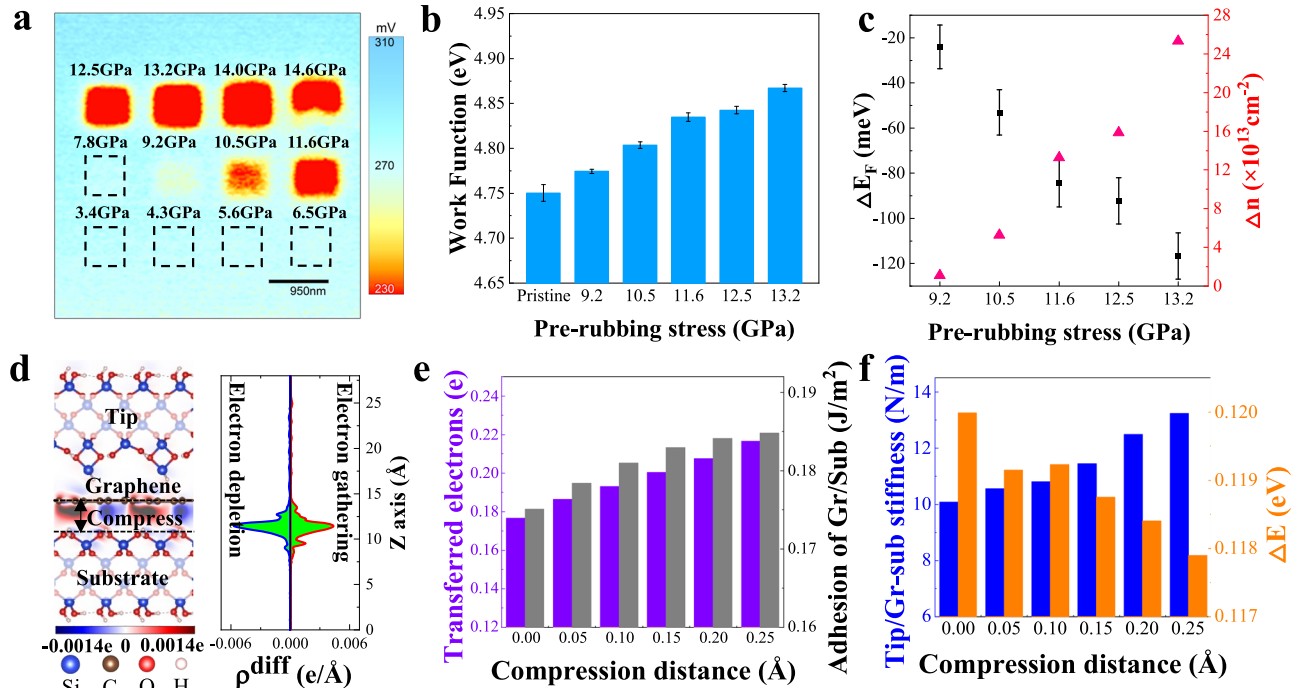

**Fig. 4 | Surface potential, interface charge transfer, adhesion, and stiffness increased after pre-rubbing, resulting in a decrease in sliding potential corrugation. a** SKPM image of graphene following varied stress pre-rubbing. **b** Work function as a function of the pre-rubbing stress. **c** The change in the Fermi level and carrier concentration. The error bars in (**b**, **c**) represent the standard deviation based on 900 data points. **d** Schematic diagram of differential charge density at the interface between graphene and substrate. The blue color indicates electron depletion, and the red color indicates electron gathering. The distance between the graphene/substrate interface will be compressed, corresponding to the pre-rubbing of the experiment. The blue, brown, red, and white atoms represent Si, C, O, and H atoms, respectively. **e** The electron transfer number and adhesion work of the graphene/substrate interface with different compression distances. **f** The tip/Gr-substrate stiffness and the sliding potential corrugation (ΔE) of the tip on the graphene under different compression distances. Source data of (**b**, **c**, **e**, **f**) are provided as a Source Data file.

rate of 1 Hz, moving from bottom to top, and then returning without detachment. Friction mapping was conducted after pre-rubbing with a Si tip (Multi75Al-G, 3.0 N/m, Budget Sensors) at a normal load of 50 nN, and the estimate of friction coefficient was carried out 10 days after the pre-rubbing process. Friction revivification experiments were conducted using an FM-LC tip. The normal and lateral cantilever sensitivities of the Multi75Al-G probe were calibrated using a noncontact method[25]. For the FM-LC probe, the normal cantilever sensitivity was also calibrated using a noncontact method, but the lateral cantilever sensitivity was calibrated using the wedge method[26] because its torsional resonance peak is outside the detection frequency range of our AFM equipment. The surface potential of graphene on SiO₂/Si substrate was done in SKPM mode after modification with a Pt-coated AFM probe (Multi75E-G, 3.0 N/m, Budget Sensors), while SiO₂ was electrically grounded during measurement, the work function of the probe was calibrated using surface potential mapping of few-layer graphene because the work function of few-layer graphene can be approximately equal to that of bulk graphite(4.65 eV)[27]. All experiments were conducted under the same conditions, maintaining a temperature of 25 °C ~ 28 °C and relative humidity of 12% ~ 15%.

**Estimation of the stress**

The relationship between the contact radius and load can be described by the Derjaguin–Müller–Toporov (DMT) model[28]:

$$a = \left[\frac{R}{K}(L + 2\pi\gamma R)\right]^{1/3} \quad (3)$$

where $a$ represents the contact radius under the load of $L$, $\gamma$ refers to the work of adhesion, and $R$ denotes the tip radius, characterized using

SEM (HITACHI S4800) after completing the pre-rubbing experiments (Supplementary Fig. 6, supporting information). $K$ stands for the combined elastic modulus, which can be calculated using the following formula:

$$K = \frac{4}{3}\left(\frac{1 - v_1^2}{E_1} + \frac{1 - v_2^2}{E_2}\right)^{-1} \quad (4)$$

in which $E$ and $v$ are the elasticity moduli and Poisson ratio of the sample and tip, respectively. Given the atomic thickness of the graphene flake, the elasticity modulus and Poisson ratio of the SiO₂ ($E_1 = 73$ GPa, $v_1 = 0.17$)[29] and the diamond ($E_2 = 1142$ GPa, $v_2 = 0.07$)[30] are used as the sample and tip in the calculation.

$\gamma$ can be estimated from the equation:

$$L_c = -2\pi\gamma/R \quad (5)$$

where $L_c$ corresponding to negative critical load, can be obtained from force curves.

Hence, the stress $\sigma$ was determined as follows:

$$\sigma = \frac{L}{\pi a^2} = \frac{L}{\pi}\left[\frac{R}{K}(L + 2\pi\gamma R)\right]^{-2/3} \quad (6)$$

**Raman spectroscopy**

Raman spectra were recorded with 532 nm laser excitation using an Invia Raman microscope.

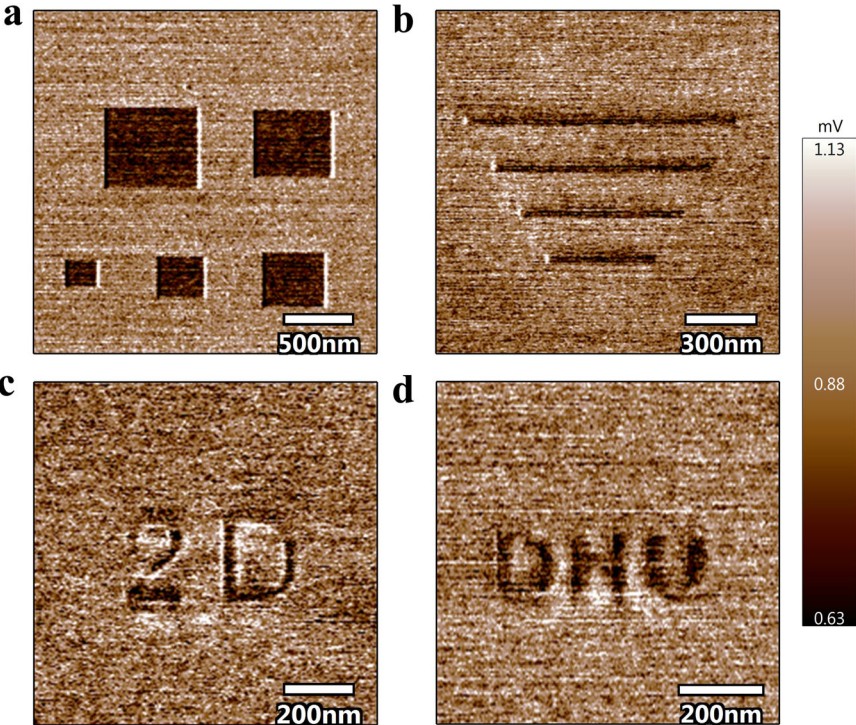

**Fig. 5 | Patterning on the surface of graphene. a–d** Showcase the intentionally created patterns including squares, lines, and selected letters, respectively.

## First-Principles Calculations

First-principles calculations based on Density Functional Theory (DFT) are performed using the Vienna Ab initio Simulation Package (VASP)[31]. The exchange–correlation function is described employing the generalized gradient approximation of Perdew–Burke–Ernzerhof[32]. The projector-augmented wave potential is utilized to represent interactions between electrons and ions. Monkhorst–Pack k–point grids are employed to sample the Brillouin zone[33]. Van der Waals (vdW) interactions are also considered in the model. The calculation parameters, including the plane–wave cutoff energy, k–points, total energy, and convergence tolerance force are set at 450 eV, 0.03 Å−1, 1×10 − 6 eV, and 0.02 eV/Å, respectively. Post-processing of the results is carried out using VASPKIT[34].

## Simulation Models

The simulation model incorporates a $SiO_2$ tip, graphene, and $SiO_2$ substrate. The interface supercell consists of AB-stacked graphite (P63/MMC) and β-cristobalite $SiO_2$ (FD-3M). The lattice parameters of graphite are a = b = 2.46 Å, c = 6.8 Å, and the lattice parameters of $SiO_2$ are a = b = c = 7.16 Å[35]. Graphene (001) surface and $SiO_2$ (001) surface are used for the calculations. The orientations of the graphene surface are [−3 1 0], [0 −3 0], and lengths are 6.53 Å and 7.40 Å, respectively. The orientations of the $SiO_2$ surface are [1 0 0], [1 0 0], and lengths are 7.06 Å and 7.06 Å, respectively. The mismatch of the two orientations is 7.8% and 4.7%, respectively. The surfaces of the tip and substrate are passivated by H atoms. The atoms within four layers near the surfaces of the tip and substrate are fixed to simulate the structure of the bulk material, while other atoms are allowed to relax. A 15 Å vacuum layer is introduced to prevent periodic interactions in all simulated supercells. The graphene on the substrate surface is compressed from the equilibrium position 0 Å to 0.2 Å to simulate the structure of the graphene after different levels of pre-rubbing. It is worth noting that one of the graphene C atoms is fixed to make graphene stay in the position after compression. The tip is compressed from the equilibrium position 0 Å to 0.4 Å to simulate the local strain of graphene caused by the tip.

Additionally, during the friction simulation, the tip slides on the graphene with a step size of 0.3 Å.

## Data availability

All data in the main text and the Supplementary Information are provided in the Source Data file. Source data are provided with this paper. Optimized DFT structures are provided as Supplementary Data 1. Source data are provided with this paper.

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

## Acknowledgements

This work was supported by the National Natural Science Foundation of China (Grant No. 52375172 (Y.P.), and 52075093 (Y.P.)), and the Shanghai Sailing Program (No. 21YF1400400 (H.L.)).

## Author contributions

H.S. designed the experiment, performed the experiments and contributed to writing the original manuscript. H.Z. conducted the DFT calculations and contributed to the original manuscript. J.S. analyzed the data and revised the manuscript. H.L. and K.Z. supervised the study and discussed the results. Y.P. supervised and directed the project. H.S., H.Z., and J.S. contributed equally to this work. All authors contributed to the writing and approval of the final manuscript.

## Competing interests

The authors declare no competing interests.
