## [Transparent Peer Review file · Nature Communications]

Reversible and Controllable Reduction in Friction of Atomically Thin Two-Dimensional Materials through High-Stress Pre-Rubbing

Corresponding Author: Professor Yitian Peng

Version 0:

Reviewer comments:

Reviewer #1

(Remarks to the Author)

Su and coworker presented the frictional characteristics of graphene layer/s in ambient conditions. The experimental study is supported by DFT simulation for charge migration and contact mechanics. The interesting concept authors bring the pre-rubbing surface to tune down the friction force and its revival under optimised pressurised rubbing. The presentation of results is clean without ambiguity, but a weak discussion is presented. Therefore, I am considering the article for acceptance after major revisions. Here are a few technical considerations for the authors.

Typo at page three.

On page 5, "crater" " graphene under deformed structure is an interesting study. Can the author figure out the possibility of strain (experimentally) through Raman spectroscopic mapping or high-resolution imaging in the graphene over deformed silica? I will recommend following article to get useful information <https://doi.org/10.1002/sml.202104487>

The SKPM technique reveals the difference in surface charge potential between pre- and post-rubbed areas. This is the critical part of the manuscript, and the author just mentioned its superficial information. The author must quantify the outcomes of SKPM following its calibration.

Three factors could be responsible for lower friction force: interfacial adhesion (value is missing), strain (possibly tension missing value) and charge accumulation (not mentioned). The author must present some analytical relation with these quantitative values.

What kind of bond formation/hybridisations/proximity effect occurred during rubbing?

Page 12: It is unclear why friction force increases if graphene is strongly adhered after pre-rubbing at lower substrate pressure.

In figure 4 (d), what is the charge distribution associated to blue color. Is it rich in electrons or holes concentration?

I am not able to find the normal and torsional stiffness of the cantilever. Please add the calibration values and its procedure.

Reviewer #2

(Remarks to the Author)

This work investigated the friction reduction of atomically thin graphene through pre-rubbing under high stress. The reduction in friction is attributed to a decrease in the maximum sliding potential barrier, induced by enhanced interactions between graphene and the substrate due to charge transfer at the interface undergoing high stress. Moreover, this reduction can be reversed by reciprocating friction under moderate stress. The results show some interesting points for optimizing and controlling the performance of atomically thin 2D materials. However, many problems still exist in the manuscript. The following issues are given for references.

1. The results show that no visible change of friction is observed at stress levels of 7.8 GPa and below. Does this stress value correspond to the threshold value leading to the charge transfer at the interface?

2. Fig. 3 shows that the friction can be reversed by reciprocating friction under moderate stresses. This suggests that the charge transfer is still within the physical change field but does not mean the formation of newly covalent bonds between the graphene and its substrate. The question is that how long time the strong interfacial interaction caused by the charge transfer can be maintained in room temperature, since a relative high temperature can cause the outstanding thermal vibrations of

atoms and relax the physically interfacial interactions.

3. If the maintaining time of the strong interfacial interactions is short, the patterning on the surface of graphene shown in Fig. 5 will have insignificant meaning for practical applications, due to the relaxations of interfaces.
4. The radius of the tip for pre-rubbing in experiment is not given.
5. More simulation details should be described, such as lattice parameters for graphene and SiO₂.
6. In DFT modelling, a SiO₂ tip is used to contact with the graphene layers attached on the surface of a SiO₂ substrate. However, in experiment procedures, the diamond tip is used to exert the pre-rubbing forces, and the silicon tip is employed to measure the friction force. The types of tip materials in experiment are clearly different from those in modelling.
7. Moreover, the SiO₂ tip and substrate in modelling have crystalline structures. However, the authors do not describe whether the substrate has crystal or amorphous lattice structures and also not give relevant evidences. This issue should be cleared, since the different lattice structure of substrates can influence the charge transfer on the contact interface.
8. The accuracy of contact stresses calculated by the DMT model should be validated.
9. Some English grammar errors are found in the manuscript. Please carefully check the writing through the whole manuscript.

Reviewer #3

(Remarks to the Author)

In this paper Su and coworkers show how the friction coefficient of graphene flakes deposited on a Si/SiO₂ can be controlled by pre-rubbing the graphene-substrate system with a diamond tip at stresses above ~8GPa.

Friction reduction is indeed clearly visible in the friction maps presented in the paper which closely match topography maps where the pre-rubbed areas appear as square-shaped craters. The Authors attribute the presence of these craters to the deformation of the SiO₂ substrate, and disregard any possible effect of the pre-rubbing procedure on graphene due to its extremely high Young's modulus.

Interestingly, this friction reduction effect can be reversed by moderate stress reciprocating friction, and moreover the surface can be decorated by nanoscale intentional patterns of modified friction areas.

The Authors explain the reduction of friction in the pre-rubbed areas with an increased interaction between graphene and the substrate leading to a strong electron transfer towards the graphene sheet and resulting in a flattened and strongly adhering graphene. Various measurements and the results of a DFT simulation presented in the paper support this claim, namely:

- (1) Atomic-scale lateral force mapping shows irregular patterns and ripples in the pristine areas, whereas, in the pre-rubbed regions, ripples are absent and the graphene lattice is visible.
- (2) The absence of the strengthening effect in the stick-slip friction behavior in the pre-rubbed regions is ascribed by the Authors to the lack of evolution of the contact quality in those areas, supporting the idea of a stronger binding of graphene to the substrate in those regions.
- (3) In the pre-rubbed area there is almost no change of friction with respect to the number of graphene layers.
- (4) DFT calculations show that, as graphene is compressed over an hydrogen-terminated SiO₂ surface, the corrugation of the potential energy surface is reduced and at the same time increasing charge is transferred at the interface.

Moreover, according to the Authors, friction reduction is not linked to changes in the substrate intrinsic properties because the bare substrate surface-potential map reveals no differences between pre-rubbed and pristine areas, and the differences in friction behavior between pristine and pre-rubbed regions are opposite with respect to the graphene covered substrate.

The results presented in the paper are, in our opinion, very interesting and worth publishing in Nature Communication, the methodology is sound and the evidence provided by Authors support their claims.

However the work would gain in significance in the field if the physical picture that the reader could grasp about what is actually happening at the interface could be deepened. This, on our opinion could be reached by considering the following:

- (1) Discuss and explain why strongly bound, flattened graphene shows less friction: indeed when graphene ripples there are local transitions of its electronic structure from sp² to sp³ configuration enhancing the reactivity of curved graphene (See P.V. Antonov, P. Restuccia, M.C. Righi, and J.W.M. Frenken, Attractive curves: the role of deformations in adhesion and friction on graphene, Nanoscale Advances 4, 4175 (2022)) these results should be discussed and properly cited.
- (2) Explain or suggest what is at the origin of the increased, reversible, adhesion between SiO₂ and graphene: is it simply due to the mechanical compression of graphene on the substrate? Is the substrate surface made more reactive by the compression? Would it be possible to deposit graphene on a pre-rubbed bare substrate and measure friction in this case? What would the Authors expect?

Other points:

(1) As the Authors write in section 2, the Young's modulus of the substrate might play a role in determining the friction behavior of graphene. They also write that "Young's modulus measurement revealed no significant changes within the pre-rubbed area". However data shown in Fig. S3(c) refers to the graphene/substrate system and not to the bare substrate, did the Authors measure the bare substrate Young's modulus map as well? Could it be shown? Or else if they refer to the graphene/substrate system, how and why does it connect to the property of the substrate alone?

(2) When discussing the DFT simulation that shows the increasing charge transfer to the interface region with the implication of greater adhesion, the paper "M. Wolloch, G. Levita, P. Restuccia, and M.C. Righi, Interfacial Charge Density and Its Connection to Adhesion and Frictional Forces, Physical Review Letters 121, 026804 (2018)" should be properly cited.

(3) More details on the calculations and justification of the model should be provided, namely: how was the common cell accommodating both the SiO₂ and graphene lattices constructed? How was strain distributed between the two? What was the reciprocal orientation? Why did the Authors choose an H termination of the SiO₂ surface? Is it representative of the experimental conditions?

Typos:

Page 3 line 7: researche → research

Page 12 "It can be deduced that the interaction between graphene and the substrate becomes stronger as the compression distance increases because the charge transferred at the interface from 0.18 e to 8.09 e" There seem to be some words missing in this sentence.

Reviewer #4

(Remarks to the Author)

Version 1:

Reviewer comments:

Reviewer #1

(Remarks to the Author)

In the revision, Peng and their coworkers satisfactorily responded to my comments. I have reviewed the revised version and found that all answers/responses have been addressed carefully. Moreover, the authors improved the discussion with clear justification. The details of SKPM, Raman spectroscopy, and simulations are technically correct and coherent with the manuscript. In my opinion, the manuscript can be accepted in the present format.

Reviewer #2

(Remarks to the Author)

The authors have addressed all my concerns in the revisions. Hence, I am happy to recommend the publication of this manuscript.

Reviewer #3

(Remarks to the Author)

The Authors replies to our comments are satisfactory, therefore we recommend this paper for publication.

Reviewer #4

(Remarks to the Author)

REVIEWER COMMENTS

Reviewer #1 (Remarks to the Author):

Su and coworker presented the frictional characteristics of graphene layer/s in ambient conditions. The experimental study is supported by DFT simulation for charge migration and contact mechanics. The interesting concept authors bring the pre-rubbing surface to tune down the friction force and its revival under optimized pressurized rubbing. The presentation of results is clean without ambiguity, but a weak discussion is presented. Therefore, I am considering the article for acceptance after major revisions. Here are a few technical considerations for the authors.

1. Typo at page three.
2. On page 5, "crater" " graphene under deformed structure is an interesting study. Can the author figure out the possibility of strain (experimentally) through Raman spectroscopic mapping or high-resolution imaging in the graphene over deformed silica? I will recommend following article to get useful information <https://doi.org/10.1002/sml.202104487>
3. The SKPM technique reveals the difference in surface charge potential between pre- and post-rubbed areas. This is the critical part of the manuscript, and the author just mentioned its superficial information. The author must quantify the outcomes of SKPM following its calibration.
4. Three factors could be responsible for lower friction force: interfacial adhesion (value is missing), strain (possibly tension missing value) and charge accumulation (not mentioned). The author must present some analytical relation with these quantitative values.
5. What kind of bond formation/hybridisations/proximity effect occurred during rubbing?
6. Page 12: It is unclear why friction force increases if graphene is strongly adhered after pre-rubbing at lower substrate pressure.
7. In figure 4 (d), what is the charge distribution associated to blue color. Is it rich in electrons or holes concentration?
8. I am not able to find the normal and torsional stiffness of the cantilever. Please add the calibration values and its procedure.

Reviewer #2 (Remarks to the Author):

This work investigated the friction reduction of atomically thin graphene through pre-rubbing under high stress. The reduction in friction is attributed to a decrease in the maximum sliding potential barrier, induced by enhanced interactions between graphene and the substrate due to charge transfer at the interface undergoing high stress. Moreover, this reduction can be reversed by reciprocating friction under moderate stress. The results show some interesting points for optimizing and controlling the performance of atomically thin 2D materials. However, many problems still exist in the manuscript. The following issues are given for references.

1. The results show that no visible change of friction is observed at stress levels of 7.8 GPa and below. Does this stress value correspond to the threshold value leading to the charge transfer at the interface?
2. Fig. 3 shows that the friction can be reversed by reciprocating friction under moderate stresses. This suggests that the charge transfer is still within the physical change field but does not mean the formation of newly covalent bonds between the graphene and its substrate. The question is that how long time the strong interfacial interaction caused by the charge transfer can be maintained in room temperature, since a relative high temperature can cause the outstanding thermal vibrations of atoms and relax the physically interfacial interactions.
3. If the maintaining time of the strong interfacial interactions is short, the patterning on the surface of graphene shown in Fig. 5 will have insignificant meaning for practical applications, due to the relaxations of interfaces.
4. The radius of the tip for pre-rubbing in experiment is not given.
5. More simulation details should be described, such as lattice parameters for graphene and SiO₂.
6. In DFT modelling, a SiO₂ tip is used to contact with the graphene layers attached on the surface of a SiO₂ substrate. However, in experiment procedures, the diamond tip is used to exert the pre-rubbing forces, and the silicon tip is employed to measure the friction force. The types of tip materials in experiment are clearly different from those in modelling.
7. Moreover, the SiO₂ tip and substrate in modelling have crystalline structures. However, the authors do not describe whether the substrate has crystal or amorphous lattice structures and also not give relevant evidences. This issue should be cleared, since the different lattice structure of substrates can influence the charge transfer on the contact interface.

8. The accuracy of contact stresses calculated by the DMT model should be validated.
9. Some English grammar errors are found in the manuscript. Please carefully check the writing through the whole manuscript.

Reviewer #3 (Remarks to the Author):

In this paper Su and coworkers show how the friction coefficient of graphene flakes deposited on a Si/SiO₂ can be controlled by pre-rubbing the graphene-substrate system with a diamond tip at stresses above ~8GPa.

Friction reduction is indeed clearly visible in the friction maps presented in the paper which closely match topography maps where the pre-rubbed areas appear as square-shaped craters. The Authors attribute the presence of these craters to the deformation of the SiO₂ substrate, and disregard any possible effect of the pre-rubbing procedure on graphene due to its extremely high Young's modulus.

Interestingly, this friction reduction effect can be reversed by moderate stress reciprocating friction, and moreover the surface can be decorated by nanoscale intentional patterns of modified friction areas.

The Authors explain the reduction of friction in the pre-rubbed areas with an increased interaction between graphene and the substrate leading to a strong electron transfer towards the graphene sheet and resulting in a flattened and strongly adhering graphene. Various measurements and the results of a DFT simulation presented in the paper support this claim, namely:

- (1) Atomic-scale lateral force mapping shows irregular patterns and ripples in the pristine areas, whereas, in the pre-rubbed regions, ripples are absent and the graphene lattice is visible.
- (2) The absence of the strengthening effect in the stick-slip friction behavior in the pre-rubbed regions is ascribed by the Authors to the lack of evolution of the contact quality in those areas, supporting the idea of a stronger binding of graphene to the substrate in those regions.
- (3) In the pre-rubbed area there is almost no change of friction with respect to the number of graphene layers.

(4) DFT calculations show that, as graphene is compressed over a hydrogen-terminated SiO₂ surface, the corrugation of the potential energy surface is reduced and at the same time increasing charge is transferred at the interface.

Moreover, according to the Authors, friction reduction is not linked to changes in the substrate intrinsic properties because the bare substrate surface-potential map reveals no differences between pre-rubbed and pristine areas, and the differences in friction behavior between pristine and pre-rubbed regions are opposite with respect to the graphene covered substrate.

The results presented in the paper are, in our opinion, very interesting and worth publishing in Nature Communication, the methodology is sound and the evidence provided by Authors support their claims.

However the work would gain in significance in the field if the physical picture that the reader could grasp about what is actually happening at the interface could be deepened. This, on our opinion could be reached by considering the following:

(1) Discuss and explain why strongly bound, flattened graphene shows less friction: indeed when graphene ripples there are local transitions of its electronic structure from sp² to sp³ configuration enhancing the reactivity of curved graphene (See P.V. Antonov, P. Restuccia, M.C. Righi, and J.W.M. Frenken, *Attractive curves: the role of deformations in adhesion and friction on graphene*, *Nanoscale Advances* 4, 4175 (2022)) these results should be discussed and properly cited.

(2) Explain or suggest what is at the origin of the increased, reversible, adhesion between SiO₂ and graphene: is it simply due to the mechanical compression of graphene on the substrate? Is the substrate surface made more reactive by the compression? Would it be possible to deposit graphene on a pre-rubbed bare substrate and measure friction in this case? What would the Authors expect?

Other points:

(1) As the Authors write in section 2, the Young's modulus of the substrate might play a role in determining the friction behavior of graphene. They also write that "Young's modulus measurement revealed no significant changes within the pre-rubbed area". However, data shown in Supplementary

Fig. 3c refers to the graphene/substrate system and not to the bare substrate, did the Authors measure the bare substrate Young's modulus map as well? Could it be shown? Or else if they refer to the graphene/substrate system, how and why does it connect to the property of the substrate alone?

(2) When discussing the DFT simulation that shows the increasing charge transfer to the interface region with the implication of greater adhesion, the paper "M. Wolloch, G. Levita, P. Restuccia, and M.C. Righi, Interfacial Charge Density and Its Connection to Adhesion and Frictional Forces, Physical Review Letters 121, 026804 (2018)" should be properly cited.

(3) More details on the calculations and justification of the model should be provided, namely: how was the common cell accommodating both the SiO₂ and graphene lattices constructed? How was strain distributed between the two? What was the reciprocal orientation? Why did the Authors choose an H termination of the SiO₂ surface? Is it representative of the experimental conditions?

Typos:

Page 3 line 7: recherche → research

Page 12 "It can be deduced that the interaction between graphene and the substrate becomes stronger as the compression distance increases because the charge transferred at the interface from 0.18 e to 8.09 e" There seem to be some words missing in this sentence.

Reviewer #4 (Remarks to the Author):

Reply to Reviewer #1

Su and coworker presented the frictional characteristics of graphene layer/s in ambient condition. The experimental study is supported by DFT simulation for charge migration and contact mechanics. The interesting concept authors bring the pre-rubbing surface to tune down the friction force and its revival under optimized pressurized rubbing. The presentation of results is clean without ambiguity, but a weak discussion is presented. Therefore, I am considering the article for acceptance after major revisions. Here are a few technical considerations for the authors.

Response: We would like to thank you for taking the time to review our manuscript. We made some careful corrections to our previous submission according to your suggestion. They largely improve the presentation of our work. Relevant corrections highlighted in blue have been added to the revised Manuscript and Supplementary information. Please find below our point-to-point responses (in blue) to your comments (in black).

1. Typo at page three.

Response: Thank you for pointing out the errors. We are sorry for the negligence. Typo on page three is corrected. For example, the sentence "Although existing researche has provided some approaches..." has been revised to "Although existing studies have provided some approaches...". We have conducted multiple reviews to eliminate the possible errors.

2. On page 5, "crater" " graphene under deformed structure is an interesting study. Can the author figure out the possibility of strain (experimentally) through Raman spectroscopic mapping or high-resolution imaging in the graphene over deformed silica? I will recommend following article to get useful information <https://doi.org/10.1002/sml.202104487>

Response: Thank you very much for the insightful comment.

The recommended article is very helpful for us to improve our manuscript and was cited as the ref. [8] in the manuscript. It investigates the interplay between texture-induced strains of graphene and its lubricated properties. The work demonstrates regulation of the frictional dissipation in nanoscale architecture through strain engineering. This study provides critical insights into assessing strain in graphene. Specifically, the Raman shifts of the G peak and 2D peak are closely correlated with strain, as variations in lattice constants result in changes to the phonon modes. The strain can be investigated

by analyzing the correlation between the frequencies of the G and 2D Raman modes of graphene (ω_G , ω_{2D}) and plotting the correlation graph according to the article mentioned above.

According to your suggestion, we also performed Raman spectroscopy mapping on graphene subjected to pre-rubbing stresses, using a Horiba HR800 Raman spectrometer with a 532 nm laser under ambient conditions. The laser power was kept below 1 mW to minimize local heating effects. The Raman spectroscopy mapping of graphene after pre-rubbing is demonstrated in Fig. R1. Fig. R1b and R1c display the intensity and position distributions of the G peak, respectively. Fig. R1d and R1e correspond to the 2D peak in the Raman mapping. An inconspicuous shift in the G or 2D peaks is observed on the graphene under a pre-rubbing stress below 14.0 GPa for friction reduction. This may indicate an inconspicuous contribution from strain to the friction reduction of the pre-rubbing graphene.

Below we evaluate the quantity of the inconspicuous strains of pre-rubbing graphene according to the method shown in the article mentioned above. Fig. R1f shows the distribution of the Raman shift for the G peak. Fig. R1g corresponds to the 2D peak. Fig. R1h depicts the correlation between the 2D and G peak positions. Each region contains 9 data points within the selected red-boxed area. The strain and doping axes are plotted with slopes of 2.2 and 0.75, respectively, corresponding to $\partial\text{Pos}2D/\partial\text{Pos}G$.¹ The range of variation in the average G frequency value is less than 1.1 cm^{-1} , while the range of variation in the average 2D frequency value is less than 1.5 cm^{-1} . The estimated tensile strain (ϵ) are as follows: $\epsilon_{\text{Pristine}} \approx \epsilon_{13.2\text{GPa}} \approx \epsilon_{14.0\text{GPa}} \approx 0.01509\%$, $\epsilon_{12.5\text{GPa}} \approx 0.01508\%$. The Raman spectroscopy indicates that after pre-rubbing under a stress of 14.0 GPa, the strain in the pre-rubbed area is so small, which may be unable to turn the friction of graphene.²

We also attempted to analyze strain using high-resolution lattice analysis. However, as shown in Fig. 1d, the irregular lattice and inherent ripples in pristine graphene make it difficult to obtain accurate comparisons.

In summary, the Raman spectroscopy results indicate that within the range of pre-rubbing stress below 13.2 GPa for friction mediation discussed in the main text, the strain is insignificant.

Figure R1. (a) AFM topography image after pre-rubbing under different stress. (b) G peak intensity map. (c) G peak frequency map. (d) G peak intensity map. (e) G peak frequency map. (f) Raman spectra of the G peak. (g) Raman spectra of the 2D peak. (h) Correlation plot of the 2D and G peak positions.

3. The SKPM technique reveals the difference in surface charge potential between pre- and post-rubbed areas. This is the critical part of the manuscript, and the author just mentioned its superficial information. The author must quantify the outcomes of SKPM following its calibration.

Response: Thank you very much for this useful comment. The questions are very interesting and deserve to be thoroughly studied for its importance in interpreting the possible mechanism of the pre-rubbing induced friction reduction. We made a more detailed discussion in the revised manuscript.

To obtain quantitative data from our KPFM measurements, the work function of the Pt-coated AFM tip (Multi75E-G, 3.0 N/m, Budget Sensors) was calibrated using surface potential mapping of few-layer graphene surface, because the work function of few-layer graphene can be approximately equal to that of bulk graphite (4.65 eV).³ The work function of the AFM tip is estimated to be 5.04 eV. This value is obtained by adding the calculated work function difference of 0.3782 eV (derived from the measured contact potential difference of 378.2 mV) to the work function of graphite. The work function of the pre-rubbed area can then be calculated using the relation:⁴

$$\Phi_{Pre} = \Phi_{tip} - eV_{CPD}$$

Where e is the electronic charge, V_{CPD} is the measured contact potential difference (CPD), Φ_{tip} is

the work function of the tip, and ϕ_{pre} is the work function of the pre-rubbed region.

Then the SKPM measurements were conducted to analyze the alterations in CPD, as illustrated in Fig. R2a-R2c.

Fig. R2a shows the SKPM images of the graphene regions pre-rubbed under the stress below 13.9 GPa and 14.6 GPa for friction modulation. Fig. R2b presents the work function. It is evident that the work function consistently increases after pre-rubbing at stress levels of 9.2 GPa and above.

The variation in surface potential can be converted into a change in the Fermi level using the following expression:⁵

$$\Delta E_F = e\Delta V_{CPD} \quad (3)$$

Where ΔV_{CPD} is the change in CPD relative to pristine graphene. Meanwhile, the change in carrier concentration in the modified graphene region can be estimated as:⁶

$$\Delta n = \frac{1}{\pi} \left(\frac{\Delta E_F}{\hbar v_F} \right)^2 \quad (4)$$

Here the \hbar is the Planck constant, v_F is the Fermi velocity. The calculated changes in the Fermi level and carrier concentration are shown in Fig. R2c. It can be observed that with the increase of pre-rubbing stress, the Fermi level of graphene gradually moves downward relative to the intrinsic level, and the carrier concentration gradually increases. This trend indicates a modification in the electronic properties of graphene due to charge transfer at the pre-rubbed graphene-substrate interface.

Further, as shown in Fig. R2d, we perform DFT calculations to investigate the interfacial charge distribution for the interfacial properties towards the pre-rubbed induced friction reduction. The results shown in Fig. R2e indicate the charge transfer between graphene and the substrate increases with indentation distance. The results are consistent with the results of SKPM measurements shown in Fig. R2c. This would lead to an increased adhesion of graphene on the substrate (Fig. R2e, right side). Accordingly, as shown in Fig. R2f, the stiffness of tip/graphene-substrate increases due to enhanced adhesion of graphene/substrate. Meanwhile, the sliding potential corrugations (Fig. R2f, right side) for the tip on graphene also decrease due to the increased interaction between the graphene and substrate.⁷ According to the PT model,⁸ the increased stiffness and decreased sliding potential corrugations would restrain friction of the tip sliding on the graphene-substrate.

Figure R2. Surface potential, interface charge transfer, adhesion, and stiffness increased after pre-rubbing, resulting in a decrease in sliding max potential corrugation. (a) SKPM image of graphene following varied stress pre-rubbing. (b) Work function as a function of the pre-rubbing stress. (c) The change in the Fermi level and carrier concentration. (d) Schematic diagram of differential charge density at the interface between graphene and substrate, the blue color indicates electron depletion, and the red color indicates electron gathering. The distance between the graphene/substrate interface will be compressed, corresponding to the pre-rubbing of the experiment. (d) The electron transfer number and adhesion work of the graphene/substrate interface with different compression distances. (e) The tip/Gr-substrate stiffness and the sliding potential corrugation (ΔE) of the tip on the graphene under different compression distances.

In summary, we have calibrated the work function of the probe and calculated the work function of the pre- and post-rubbed areas. SKPM measurements indicate the increased adhesion of graphene on the substrate with increasing pre-rubbing stress due to the charge transfer. The results of DFT calculations are consistent with the SKPM experiments. The adhesion enhancement of graphene on the substrate leads to increased stiffness and decreased sliding potential corrugations, which would restrain friction of the tip sliding on the graphene-substrate.

Revisions of the main text:

(1) We have added the SKPM measurement procedure and the calibration of the probe's work function to the Experimental Section of the main text:

“The surface potential of graphene on SiO_2/Si substrate was done in SKPM mode after modification with a Pt-coated AFM probe (Multi75E-G, 3.0 N/m, Budget Sensors), while SiO_2 was

electrically grounded during measurement, the work function of the probe was calibrated using surface potential mapping of few-layer graphene because the work function of few-layer graphene can be approximately equal to that of bulk graphite(4.65eV).³”

(2) We have included Fig. R2 in the main text as Fig. 4 and have added the corresponding discussion of Fig. R2 to the main text.

4. Three factors could be responsible for lower friction force: interfacial adhesion (value is missing), strain (possibly tension missing value) and charge accumulation (not mentioned). The author must present some analytical relation with these quantitative values.

Response: Thank you for this insightful comment. We are sorry for the omission of the discussion about the factors that may be responsible for the lower friction. If we have well understood the above comments, your concerns may be addressed as below.

(1) Interfacial adhesion of tip/graphene as well as the adhesion of graphene-substrate may be indeed a critical factor in determining friction. ① For the adhesion of tip/graphene, we conducted supplementary experiments using an FM-LC probe to apply high-stress pre-rubbing at 11.6 GPa on selected areas, followed by adhesion force mapping measurements over an expanded region to assess changes in adhesion force. As expected, shown in the main text, the friction in the pre-rubbed area significantly decreases. Subsequently, we performed adhesion force mapping over the expanded region. The results are shown in Fig. R3. Fig. R3a and R3b show the AFM morphology of the enlarged area and the friction force mapping under a 50nN load, respectively. Fig. R3e presents the corresponding adhesion force mapping. As shown in Fig. R3f, the adhesion force of tip/graphene in the pre-rubbed area did not exhibit a significant change. Statistical analysis revealed that the average adhesion force in the pre-rubbed area was reduced by only 0.6% compared to the pristine graphene. According to the equation:⁹

$$F = \tau \pi a$$

and the equation derived from the DMT model:¹⁰

$$a = \left[\frac{R}{K} (L - L_c R^2) \right]^{1/3}$$

Where τ is the interfacial shear strength. a represents the contact radius under the load of L , and R denotes the tip radius, L_c corresponding to the adhesion force between the tip and graphene. The

decrease in adhesive force L_c results in a frictional reduction of 0.2%. Therefore, the change in graphene-tip adhesion is not the primary cause of the reduced friction in the pre-rubbed region.

② We further calculated the adhesion work between graphene and the substrate under different compression distances by DFT, because of the challenge in experimental measurement. As shown in Fig. R3g. The adhesion work of the graphene-substrate increases by 0.01 J/m^2 with compression, indicating an increase in the adhesive strength between the graphene and the substrate. As stated in reply for Question 3 above, this leads to increased stiffness and decreased sliding potential corrugations, as shown in Fig. R3h and R3i, which would restrain friction of the tip sliding on the graphene-substrate.

Figure R3. (a) Morphology of the area containing the pre-rubbed region after high-stress pre-rubbing. (b) Friction force map obtained simultaneously with the morphology. (c) Comparison of friction force between the pre-rubbed region and the pristine graphene. (d) Height data during the adhesion force

mapping process. (e) Adhesion force mapping. (f) Comparison of adhesion force between the pre-rubbed graphene and the pristine graphene. (g) Adhesion work between graphene and the substrate with different compression distances. (h, i) The increased stiffness and decreased sliding potential corrugations (ΔE) with different compression distances.

(2) As stated in reply for Question 2 for the strain, Raman spectroscopy results show no significant shifts in the characteristic peaks within the range of friction reduction explored in the main text (with pre-rubbing stress below 13.2 GPa).

Furthermore, a series of pre-rubbing cycling experiments further indicate that the reduction of friction in the pre-rubbed region may not be primarily dominated by strain. as shown in Fig. R4. The experiment involved comparing friction forces after different numbers of pre-rubbing cycles at 11.6 GPa. After 1, 2, 4, and 8 pre-rubbing cycles, friction tests were conducted over an expanded area. Fig. R4a shows the morphology after pre-rubbing, and Fig. R4b shows the friction force mapping at a load of 50 nN.

Figure R4. (a) The morphology of areas subjected to varying numbers of pre-rubbing cycles. (b) The corresponding friction mapping. (c) The calculated strain and friction forces as a function of different cycles of pre-rubbing cycles.

Then, assuming that all the graphene within the “crater” after pre-rubbing results from deformation, the change in the area of the graphene following crater formation can be calculated using the equation $\Delta A = 4\delta l$, where δ represents the sinking height and l represents the side length of the pre-rubbed area. Thus, the strain can be calculated using the equation $\varepsilon = 4\delta l/l^2 = 4\delta/l$. As shown in Fig. R4c, with the increase in pre-rubbing cycles, the calculated strain increased significantly, from 0.652% to 1.20%, but the change in friction was minimal. The friction force remained almost the same after different numbers of pre-rubbing cycles. Therefore, we conclude that the changes in friction force in the pre-rubbed regions may not be primarily driven by strain.

(3) As stated in reply for Question 3 for the charge accumulation, the charge transfer between graphene and substrate enhances the adhesion of graphene on the substrate, which leads to increased stiffness and decreased sliding potential corrugations for friction reduction.

5. What kind of bond formation/hybridisations/proximity effect occurred during rubbing?

Response: Thank you for this comment. The density of states and crystal orbital Hamiltonian population (COHP) was used to characterize the effect of bond formation/hybridisations/proximity in Fig. R5.

(1) As shown in Fig. R5a, there was no significant change in the density of states of the system, indicating few bonds formation or breaking during the compression distance 0\AA , 0.1\AA , and 0.2\AA . The interaction strength between graphene and substrate can be characterized by COHP. -COHP greater than 0 is a bonding electron, and COHP less than 0 is an antibonding electron. Comparing Fig. R5b and Fig. R5c, the -COHP (0.22) of graphene/substrate much is less than the -COHP (10) of C-C bond in the graphene. Thus, there is a weak interaction instead of the bond formation at the graphene/substrate interface. The Raman spectroscopy results show no significant shift in the G peak position and no appearance of the D peak (Fig. R1) in the pre-rubbed regions below 14 GPa, indicating that the in-plane carbon-carbon vibration modes of graphene remain unchanged and that no defects, edges, or lattice distortions have formed in the graphene. This further suggests that no new bond formation has occurred.

In summary, DFT calculations and Raman spectroscopy indicate that there is no bond formation at the graphene/substrate interface.

(2) In Fig. R5b, more electrons are transferred to lower energy levels between -23 eV and -18 eV, and there is an increase in bonding electrons between -10 eV and -5 eV when the compression distance is 0.2\AA . This is attributed to the orbital hybridization effect between graphene and the substrate when the distance is compressed, and more bonding orbitals is formed that are occupied by electrons.

In summary, DFT calculations indicate that an orbital hybridization effect occurs in the pre-rubbed region, resulting in the formation of additional bonding orbitals that are occupied by electrons.

Figure R5. Density of states and the crystal orbitals Hamiltonian population (COHP) during rubbing. (a) Electronic density of states of the tip/graphene/substrate interface models with compression distances 0 Å, 0.1 Å, and 0.2 Å. (b) is crystal orbitals Hamiltonian population which indicates the interaction strength of the atom between graphene and substrate. (c) is crystal orbitals Hamiltonian population of C-C bond in the graphene.

(3) In the context of our work, the "proximity effect" refers to the reduction in the distance between the graphene layer and the substrate during pre-rubbing. This closer proximity facilitates stronger electronic interactions between the graphene and substrate, promoting charge transfer across the interface. This charge transfer contributes to the enhanced interfacial adhesion, as evidenced by the DFT calculations.

Moreover, the proximity effect in our system also implies that the increased adhesion is not solely due to mechanical deformation but is driven by the electronic coupling at the interface, as a result of the reduced graphene-substrate separation. This coupling alters the local electronic properties, such as the Fermi level, and helps to explain the observed changes in surface potential, as confirmed by our SKPM measurements.

Revisions of the main text:

We have added the analysis of Crystal Orbital Hamiltonian Population (COHP) data to Page 12 of the main text:

“The interaction strength between the graphene and substrate atoms is also discussed by crystal orbital Hamiltonian population (COHP) shown in Supplementary Fig. 5. It also implied that pre-rubbing leads to increased adhesion of graphene on the substrate.”

Revisions of the Supplementary information: We have included Fig. R4 in the Supplementary information as Supplementary Fig. 5.

6. Page 12: It is unclear why friction force increases if graphene is strongly adhered after pre-rubbing at lower substrate pressure.

Response: Thank you for this comment. We are sorry for the vague expression in the previous submission. Our description and discussion of the friction revivification experiments were not detailed enough.

For clarity, we will use "Rub" to refer to the process under high and moderate stress, and "Friction" to refer to friction force changes occurring entirely under low stress. The friction behavior in the pre-rubbed region proceeds as follows: After high-stress pre-rubbing, friction in the pre-rubbed area decreases significantly due to charge transfer, which enhances adhesion between the graphene and the substrate, lowering the maximum potential corrugation. However, after subsequent moderate-stress reciprocating rubbing, friction in the pre-rubbed area increases, eventually returning to a level consistent with pristine graphene. This occurs because the adhesion strength between the graphene and the substrate reverts to its original state after moderate-stress reciprocating rubbing, causing the friction force to similarly return to the level of the surrounding pristine graphene.

The reversion of adhesion strength occurs because, during the reciprocating rubbing under moderate stress, local stress is introduced to the graphene at the bottom edge of the "crater," as shown in Fig. R6. The tightly bonded regions gradually begin to detach from the edges and extend across the entire region. As a result, the separated areas exhibit frictional forces that revert to match those of the pristine graphene, consistent with the friction recovery process shown in Fig. 3a. Additional reciprocating rubbing experiments were performed, first at 1.58 GPa within an enlarged region that included the pre-rubbed area, and second at 5.6 GPa within the pre-rubbed area itself. These experiments demonstrated that the pre-rubbed region can sustain a low-friction state. This indicates that the variation in adhesion strength indeed begins from the edge of the "crater".

Figure R6. Illustration of the different stages of interfacial adhesion strength. (a) Pristine graphene is deposited on the substrate with weak adhesion. (b) Enhanced interfacial adhesion after high-stress pre-rubbing. (c) Detachment of strongly adhered regions induced by local stress; the enlarged image shows the changes in the edge of the pre-rubbed region under moderate-stress rubbing. (d) The interfacial adhesion strength in the pre-rubbed region fully recovers to match that of the pristine graphene.

7. In figure 4 (d), what is the charge distribution associated to blue color. Is it rich in electrons or holes concentration?

Response: Thank you for the reminder. We are sorry for the vague expression on charge distribution in the previous submission. We have discussed the transfer of interface charges and their effects in Fig. R7.

As shown in Fig. R7a, the schematic diagram of differential charge density calculation. It is that the charge density of tip/graphene/substrate subtracts the charge density of substrate and tip/graphene. The blue color represents electron depletion and the red color represents electrons gathering. Fig R7b-R7d show the differential charge density at the interface between graphene and substrate with compression distance (Comp.) 0 Å, 0.1 Å, and 0.2 Å. It can be observed that the transfer of interface charges increases with compression distance increase.

Figure R7. (a) Schematic diagram of differential charge density calculation. The blue color represents electron depletion and the red color represents electron gathering. (b)-(d) Differential charge density at the interface between graphene and substrate with compression distance (Comp.) 0 Å, 0.1 Å, and 0.2 Å.

Revisions of the main text: We have reorganized Fig. 4 of the main text to discuss how interfacial

adhesion affects for the sliding max potential corrugation and friction of graphene.

8. I am not able to find the normal and torsional stiffness of the cantilever. Please add the calibration values and its procedure.

Response: Thank you for this comment. The principle of normal and lateral cantilever sensitivities of the Multi75Al-G probe was calibrated based on a noncontact method.¹¹ Specifically, we obtain the thermal noise spectrum of the probe based on the parameters provided by the probe supplier, and the calibration can be automatically performed using the embedded program in our AFM equipment. For the FM-LC probe, the normal cantilever sensitivity was also calibrated using a noncontact method, but the lateral cantilever sensitivity was calibrated using the wedge method¹² because its torsional resonance peak is outside the detection frequency range of our AFM equipment. Specifically, we scan the probe across a sloped surface at varied loads. The lateral sensitivity is then derived based on the friction loop's response under different normal loads by considering the cantilever's geometry. Calibration values are shown in Table R1.

Table R1. Cantilever sensitivity calibration values of the probes.

	Multi75Al-G	FM-LC
Normal sensitivities	177nN/V	1150nN/V
Lateral sensitivities	535nN/V	3810nN/V

Revisions of the main text: We have added the probe calibration procedure to the experimental section of the main text:

“The normal and lateral cantilever sensitivities of the Multi75Al-G probe were calibrated using a noncontact method¹¹. For the FM-LC probe, the normal cantilever sensitivity was also calibrated using a noncontact method, but the lateral cantilever sensitivity was calibrated using the wedge method¹² because its torsional resonance peak is outside the detection frequency range of our AFM equipment.”

Reply to Reviewer #2

This work investigated the friction reduction of atomically thin graphene through pre-rubbing under high stress. The reduction in friction is attributed to a decrease in the maximum sliding potential barrier, induced by enhanced interactions between graphene and the substrate due to charge transfer at the interface undergoing high stress. Moreover, this reduction can be reversed by reciprocating friction under moderate stress. The results show some interesting points for optimizing and controlling the performance of atomically thin 2D materials. However, many problems still exist in the manuscript. The following issues are given for references.

Response: We would like to thank you for taking the time to review our manuscript. Your suggestion largely improves the presentation of our work. Please find below our point-to-point responses (in blue) to your comments (in black).

1. The results show that no visible change of friction is observed at stress levels of 7.8 GPa and below. Does this stress value correspond to the threshold value leading to the charge transfer at the interface?

Response: Thank you for this insightful comment.

Yes. There is no visible change of friction at stress levels of 7.8 GPa and below. Strictly, it is hard to say this stress value correspond to the threshold value leading to the charge transfer at the interface. Also, it can be deduced that this stress value almost corresponds to the threshold value leading to the strong charge transfer to enhance the adhesion at the interface. The charge transfer between graphene and the substrate occurs and increases with indentation distance correlating with the stress according our DFT calculations, as shown in Fig. R8b. This stress value ~ 7.8 GPa may correspond to the threshold value to strong charge transfer to enhance the adhesion at the interface, then reduce the friction.

Figure R8. (a) Schematic diagram of differential charge density at the interface between graphene and substrate. The blue color indicates electron depletion, and the red color indicates electron gathering. The distance between the graphene/substrate interface will be compressed, corresponding to the pre-rubbing of the experiment. (b) The electron transfer number and adhesion work of the graphene/substrate interface with different compression distances.

In summary, this stress value ~ 7.8 GPa almost correspond to the threshold value leading to the strong charge transfer to enhance the adhesion at the interface and decrease the friction. The threshold value ~ 7.8 GPa corresponds to the threshold value leading to the reduction of friction.

2. Fig. 3 shows that the friction can be reversed by reciprocating friction under moderate stresses. This suggests that the charge transfer is still within the physical change field but does not mean the formation of newly covalent bonds between the graphene and its substrate. The question is that how long time the strong interfacial interaction caused by the charge transfer can be maintained in room temperature, since a relative high temperature can cause the outstanding thermal vibrations of atoms and relax the physically interfacial interactions.

Response: Thank you for this insightful comment. We had conducted friction tests on the patterned area using an FM-LC probe at 50 nN both immediately after pre-rubbing and 10 days later. The samples had been stored under environmental conditions, with temperature variations between 25.4 and 32.1 °C. Fig. R9 a and b show the friction maps, while Fig. R9c presents a comparison of friction between the pristine region and the pre-rubbed region at different time points. Due to the passivation of the probe, the friction data was normalized to facilitate comparison. The normalized friction of the pre-rubbed region slightly increased after 10 days of storage, as you suggested, likely due to enhanced atomic thermal vibrations and the relaxation of physical interfacial interactions. However, it is evident that the reduction in friction caused by pre-rubbing had remained relatively low even after several days.

Figure R9. Friction maps of patterned graphene: (a) after pre-rubbing and (b) 10 days post pre-rubbing. (c) Comparison of friction between the pre-rubbing area and the pristine area at various time points.

In summary, the friction reduction from the strong interfacial interaction caused by the charge transfer can be maintained stable after 10 days storage at room temperature.

3. If the maintaining time of the strong interfacial interactions is short, the patterning on the surface of graphene shown in Fig. 5 will have insignificant meaning for practical applications, due to the relaxations of interfaces.

Response: Thank you for this insightful comment. As stated in reply for Question 2, the friction reduction from the strong interfacial interaction caused by the charge transfer can be maintained stable after 10 days storage at room temperature.

4. The radius of the tip for pre-rubbing in experiment is not given.

Response: Thank you for this valuable comment. The radius of the diamond tip (FM-LC) was characterized using SEM (HITACHI S4800) after completing the pre-rubbing experiments and was estimated to be 17 nm, as shown in Fig. R10.

Figure R10. SEM images of the probe and an estimation of the tip radius after pre-rubbing.

Revisions of the Supplementary information: We have included Fig. R10 in the Supplementary information as Supplementary Fig. 6.

5. More simulation details should be described, such as lattice parameters for graphene and SiO₂.

Response: Thank you for this insightful comment. We have added more details on establishing computational models in the methods section.

The interface supercell consists of AB-stacked graphite (P63/MMC) and β -cristobalite SiO₂ (FD-3M). The lattice parameters of graphene are $a=b=2.46 \text{ \AA}$, $c=6.8 \text{ \AA}$, and the lattice parameters of SiO₂ are $a=b=c=7.16 \text{ \AA}$ ^{13, 14}. Graphene (001) surface and SiO₂ (001) surface are used for the calculations. The orientations of the graphene surface are $[-3 \ 1 \ 0]$, $[0 \ -3 \ 0]$, and lengths are 6.53 \AA and 7.40 \AA , respectively. The orientations of the SiO₂ surface are $[1 \ 0 \ 0]$, $[1 \ 0 \ 0]$, and lengths are 7.06 \AA and 7.06 \AA , respectively. The mismatch of the two orientations is 7.8% and 4.7%, respectively.

6. In DFT modelling, a SiO₂ tip is used to contact with the graphene layers attached on the surface of a SiO₂ substrate. However, in experiment procedures, the diamond tip is used to exert the pre-rubbing forces, and the silicon tip is employed to measure the friction force. The types of tip materials in experiment are clearly different from those in modelling.

Response: Thank you for this insightful comment. We are sorry that the explanation of the models is unclear, leading to some misunderstanding.

As your understanding, in AFM experiments, the silicon tip is used to measure friction force, not to exert the pre-rubbing forces. The SiO₂ tip is used to measure friction in DFT, meanwhile, the pre-rubbing is achieved by adjusting the distance between the graphene and the substrate. The Si tip is usually oxidized to SiO₂ in the atmospheric environment¹⁵. Therefore, in the DFT calculation, the SiO₂ tip model is used instead of the Si.

7. Moreover, the SiO₂ tip and substrate in modelling have crystalline structures. However, the authors do not describe whether the substrate has crystal or amorphous lattice structures and also not give relevant evidences. This issue should be cleared, since the different lattice structure of substrates can influence the charge transfer on the contact interface.

Response: Thank you for this insightful comment. We strongly agree that different lattice structures

of substrates can influence the charge transfer on the contact interface.

In general, the substrate used in this work is a silicon wafer with a 300nm thick SiO₂ layer, prepared through thermal oxidation, and the structure of the SiO₂ formed by this process could be different structures, such as quartz, cristobalite, and amorphous structures^{16, 17}. These different structures may lead to the different of surface charge transfer and chemical activity. However, the interfacial charge transfer after pre-rubbing, enhance the adhesion between the substrate and graphene to reduce the friction. It is consistent with the trend in the manuscript. The interfacial charge transfer and adhesion are mainly enhanced by the distance interface of graphene/substrate that corresponding to the stress. We also conduct the experiment to pre-rubbing on the surface of the MoS₂ on SiO₂. The same phenomena that the friction on the MoS₂ reduced after pre-rubbing, as shown in Fig. R11.

Secondly, for the computational expense, the SiO₂ crystalline structures are often used as models for amorphous silica surfaces^{18, 19}. On the one hand, it is usually assumed within the context of this model that the surfaces of amorphous silica grains are composed of different SiO₂ surfaces (especially the 001 and 111 surfaces)²⁰. On the other hand, the size of supercells is usually about 10 or 20 angstroms. The model is still ordered in the long range due to the periodicity considered in the DFT calculation, making it difficult to simulate the long-range disordered structure of amorphous materials in experiments. Therefore, crystalline structures are acceptable for the simulation in the DFT models.

Figure R11. Pre-rubbing and friction measurements of MoS₂ on SiO₂/Si substrate. (a) Optical image of a 1.1nm thick MoS₂ flake deposited on SiO₂/Si substrate. (b) Topography image after pre-rubbing under 10.0 GPa, note that a drift occurs during the scanning-down stage. (c) Friction mapping after pre-rubbing (with a normal load of 50nN) acquired simultaneously with topography.

8. The accuracy of contact stresses calculated by the DMT model should be validated.

Response: Thank you for this insightful comment. The reason that we utilized the DMT model to estimate the contact stress is that, when surface forces are short-ranged relative to the elastic deformations they induce, the DMT model provides an effective estimation of the contact radius,

particularly for rigid materials, weak adhesive forces, and small tip radii²¹.

In addition, the Hertzian model is also used to estimate contact areas at the nanoscale. In a previous study, researchers conducted ultra-high load experiments with a diamond tip on graphene deposited on a SiO₂ substrate, similar to our setup, and estimated the contact stress using the Hertzian model²². Therefore, we also applied the Hertzian model for calculating the contact stress as a reference.

The contact radius a can be determined following the Hertzian model²³:

$$a = \sqrt[3]{\frac{3FR}{4E^*}}$$

Where F represents the applied force, R denotes the tip radius, and E^* stands for the effective elastic modulus, which is calculated using the following formula:

$$\frac{1}{E^*} = \frac{1 - \nu_{tip}^2}{E_{tip}} + \frac{1 - \nu_{sample}^2}{E_{sample}}$$

in which E and ν are the elasticity modulus and Poisson ratio, respectively.

The mean contact stress σ exerted on the sample by the tip was determined as follows:

$$\sigma = \frac{F}{\pi a^2} = \frac{1}{\pi} \left(\frac{4E^*}{3} \right)^{2/3} \left(\frac{F}{R^2} \right)^{1/3}$$

The contact stress estimated using both the DMT and Hertz models is presented in Table R2.

Table R2. Contact stress estimated through different models.

Normal load (nN)	60	100	200	300	500	800	1200	1600	2000	2400
Contact stress estimated through DMT model (GPa)	3.4	4.3	5.6	6.5	7.8	9.2	10.5	11.6	12.5	13.2
Contact stress estimated through Hertzian model (GPa)	3.9	4.6	5.8	6.7	7.9	9.3	10.6	11.7	12.6	13.3

Based on Table R2, it can be observed that at low loads, the contact stress estimated using the Hertzian model is higher. However, as the load increases, the contact stress estimated by both models converges, particularly in the friction mediation range discussed in the main text, where the contact stress becomes largely consistent.

In summary, the contact stress estimated through the DMT model can be considered accurate.

9. Some English grammar errors are found in the manuscript. Please carefully check the writing through the whole manuscript.

Response: Thank you for this valuable and thoughtful comment. We are sorry for the negligence. We have carefully reviewed and improved the English writing in the revised manuscript.

Reply to Reviewer #3

In this paper Su and coworkers show how the friction coefficient of graphene flakes deposited on a Si/SiO₂ can be controlled by pre-rubbing the graphene-substrate system with a diamond tip at stresses above ~8GPa.

Friction reduction is indeed clearly visible in the friction maps presented in the paper which closely match topography maps where the pre-rubbed areas appear as square-shaped craters. The Authors attribute the presence of these craters to the deformation of the SiO₂ substrate, and disregard any possible effect of the pre-rubbing procedure on graphene due to its extremely high Young's modulus.

Interestingly, this friction reduction effect can be reversed by moderate stress reciprocating friction, and moreover the surface can be decorated by nanoscale intentional patterns of modified friction areas.

The Authors explain the reduction of friction in the pre-rubbed areas with an increased interaction between graphene and the substrate leading to a strong electron transfer towards the graphene sheet and resulting in a flattened and strongly adhering graphene. Various measurements and the results of a DFT simulation presented in the paper support this claim, namely:

- (1) Atomic-scale lateral force mapping shows irregular patterns and ripples in the pristine areas, whereas, in the pre-rubbed regions, ripples are absent and the graphene lattice is visible.
- (2) The absence of the strengthening effect in the stick-slip friction behavior in the pre-rubbed regions is ascribed by the Authors to the lack of evolution of the contact quality in those areas, supporting the idea of a stronger binding of graphene to the substrate in those regions.
- (3) In the pre-rubbed area there is almost no change of friction with respect to the number of graphene layers.
- (4) DFT calculations show that, as graphene is compressed over a hydrogen-terminated SiO₂ surface, the corrugation of the potential energy surface is reduced and at the same time increasing charge is transferred at the interface.

Moreover, according to the Authors, friction reduction is not linked to changes in the substrate

intrinsic properties because the bare substrate surface-potential map reveals no differences between pre-rubbed and pristine areas, and the differences in friction behavior between pristine and pre-rubbed regions are opposite with respect to the graphene covered substrate.

The results presented in the paper are, in our opinion, very interesting and worth publishing in Nature Communication, the methodology is sound and the evidence provided by Authors support their claims.

However, the work would gain in significance in the field if the physical picture that the reader could grasp about what is actually happening at the interface could be deepened. This, on our opinion could be reached by considering the following:

Response: We would like to thank you for taking the time to review our manuscript. Your suggestion largely improves the presentation of our work. Please find below our point-to-point responses (in blue) to your comments (in black).

(1) Discuss and explain why strongly bound, flattened graphene shows less friction: indeed, when graphene ripples there are local transitions of its electronic structure from sp² to sp³ configuration enhancing the reactivity of curved graphene (See P.V. Antonov, P. Restuccia, M.C. Righi, and J.W.M. Frenken, Attractive curves: the role of deformations in adhesion and friction on graphene, Nanoscale Advances 4, 4175 (2022)) these results should be discussed and properly cited.

Response: Thank you very much for the insightful comment. This recommended article demonstrates that the greater the deformation of graphene, the higher its activity because of the increased charge transfer (sp² change to sp³), leading to increased friction. The recommended article is very helpful for us to improve our manuscript and was cited as the ref. [19] in the manuscript.

As shown in Fig. R12, the z-direction deformation (Fig. R12 a-c) of graphene decreases with compression distance 0 Å, 0.1 Å, 0.2Å when the tip contacts the graphene. The adhesion (Fig. R12 d) between the graphene and the substrate increases because the interface charge increases²⁴. Similar to the recommended article, the surface activity of graphene decreases with graphene ripples decreases, due to the adhesion of the interface graphene/substrate enhancement. Therefore, the adhesion between the tip and graphene decreases (Fig. R12 e), and the sliding potential barriers (ΔE) decrease (Fig. R12 f). We have cited the paper in the sentence “the stiffness of tip/graphene-substrate

increases due to enhanced adhesion of graphene/substrate. The reactivity of graphene decreases due to its decreased deformation” on page 12 of the main text.

Figure R12. The influence of z-direction deformation of graphene on tip/graphene adhesion and sliding potential barrier (ΔE). a-c are the z-direction deformation of graphene with compression distance 0 Å, 0.1 Å, 0.2 Å when the tip contacts the graphene. d is adhesion and the number of transferred electrons of the graphene/substance interface with different compression distances. e and f are the adhesion and sliding potential barriers of the tip/graphene interface.

(2) Explain or suggest what is at the origin of the increased, reversible, adhesion between SiO_2 and graphene: is it simply due to the mechanical compression of graphene on the substrate? Is the substrate surface made more reactive by the compression? Would it be possible to deposit graphene on a pre-rubbed bare substrate and measure friction in this case? What would the Authors expect?

Response: Thank you for this insightful comment. The questions deserve to be thoroughly studied for its importance in interpreting the possible mechanism of the pre-rubbing induced friction reduction.

① The increased adhesion between SiO_2 and graphene is attributed to the interfacial charge transfer subjected to high-stress pre-rubbing.

The Scanning Kelvin Probe Microscopy (SKPM) was conducted to analyze the changes at the interface. Fig. R13 (a) shows the SKPM images of the graphene regions pre-rubbed under the stress below at 13.9 GPa and 14.6 GPa for friction modulation, and Fig. R13 (b) presents the surface work

function. It is evident that the work function increases after pre-rubbing at stress levels of 9.2 GPa and above.

The variation in contact potential difference can be converted into a change in the Fermi level using the following expression:⁵

$$\Delta E_F = e\Delta V_{CPD}$$

where e is the elementary charge, and ΔV_{CPD} is the change in CPD relative to pristine graphene. Meanwhile, the change in carrier concentration in the modified graphene region can be estimated as:⁶

$$\Delta n = \frac{1}{\pi} \left(\frac{\Delta E_F}{\hbar v_F} \right)^2$$

Here the \hbar is the Planck constant, and v_F is the Fermi velocity. The calculated changes in the Fermi level and carrier concentration are shown in Fig. R13 (c). It can be observed that the Fermi level of graphene gradually shifts downward with increasing pre-rubbing stress, and the carrier concentration gradually increases. The results of DFT calculations also demonstrate that pre-rubbing leads to a decrease in the Fermi level of graphene/substrate, as shown in Supplementary Fig. R13.

The DFT calculations was performed to investigate the interfacial charge distribution, adhesion strength of graphene on substrate, stiffness, and the sliding energy barrier for the tip on the graphene-substrate towards the pre-rubbed induced friction reduction. The results shown in Fig. R13 (e) indicate the charge transfer between graphene and the substrate increases with indentation (compress) distance. The results are consistent with the results of SKPM measurements shown in Fig. R13 (c), because the charge transfer at the graphene-substrate interface generally causes doping of the graphene layer.²⁵ This would lead to an increased adhesion of graphene on the substrate (Fig. R13 (e), right side). The interaction strength between the graphene and substrate atoms is also discussed by the crystal orbital Hamiltonian population (COHP) shown in Supplementary Fig. R13 (g). It also implied that pre-rubbing leads to increased adhesion of graphene on the substrate. Further, as shown in Fig. R13 (f), the stiffness of tip/graphene-substrate increases due to enhanced adhesion of graphene/substrate. Meanwhile, the sliding potential corrugations (Fig. R13(f), right side) for tip on graphene also decrease due to the increased interaction between the graphene and substrate.⁷ According to the PT model,⁸ the increased stiffness and decreased sliding potential corrugations would restrain friction of the tip sliding on the graphene-substrate.

Figure R13. Surface potential, interface charge transfer, adhesion, and stiffness increased after pre-rubbing, resulting in a decrease in sliding potential corrugation. (a) SKPM image of graphene following varied stress pre-rubbing. (b) Work function as a function of the pre-rubbing stress. (c) The change in the Fermi level and carrier concentration. (d) Schematic diagram of differential charge density at the interface between graphene and substrate. The blue color indicates electron depletion, and the red color indicates electron gathering. The distance between the graphene/substrate interface will be compressed, corresponding to the pre-rubbing of the experiment. (e) The electron transfer number and adhesion work of the graphene/substrate interface with different compression distances. (f) The tip/Gr-substrate stiffness and the sliding potential corrugation (ΔE) of the tip on the graphene under different compression distances. (g-h) Crystal orbitals Hamiltonian population (COHP) under different compression conditions, (i) COHP integral diagram under different compression distance.

② The reversibility of the adhesion between SiO_2 and graphene is attributed to the introduction of local stress to the graphene at the bottom edge of the "crater" during reciprocating rubbing. As a result, the tightly bonded regions gradually begin to detach from the edges and extend across the entire area. As shown in Fig R14. As a result, the separated areas exhibit frictional forces that revert

to match those of the pristine graphene, consistent with the friction recovery process shown in Fig. 3a.

Figure R14. Illustration of the different stages of interfacial adhesion strength. (a) Pristine graphene is deposited on the substrate with weak adhesion. (b) Enhanced interfacial adhesion after high-stress pre-rubbing. (c) Detachment of strongly adhered regions induced by local stress; the enlarged image shows the changes in the edge of the pre-rubbed region under moderate-stress rubbing. (d) The interfacial adhesion strength in the pre-rubbed region fully recovers to match that of the pristine graphene.

③ In our DFT calculations, the enhancement of adhesion occurs by compressing from a non-optimal interfacial spacing to an optimal one, with no significant changes in the electronic structure, as shown in Fig. R13 g. Therefore, we believe that the reactivity of the substrate does not show notable variations.

④ Depositing atomically thin graphene onto a pre-rubbed bare substrate poses a significant challenge. We made attempts, but the attempts failed. We found that the most daunting aspect was accurately transferring atomically thin graphene, which is nearly transparent and on the micrometer scale, to the pre-rubbed area of bare SiO₂, also of micrometer size and not easily locatable under an optical microscope.

Other points:

(1) As the Authors write in section 2, the Young's modulus of the substrate might play a role in determining the friction behavior of graphene. They also write that "Young's modulus measurement revealed no significant changes within the pre-rubbed area". However, data shown in Fig. S3(c) refers to the graphene/substrate system and not to the bare substrate, did the Authors measure the bare substrate Young's modulus map as well? Could it be shown? Or else if they refer to the graphene/substrate system, how and why does it connect to the property of the substrate alone?

Response: Thank you for this insightful comment. We apologize for the ambiguity caused by our previous statements. We had not measured the Young's modulus of the bare substrate after pre-rubbing. Our intention was to refer to the lack of change in the Young's modulus of the graphene/substrate system after pre-rubbing, indicating that the reduction in friction is not due to a change in Young's modulus of the graphene/substrate system. However, our previous statements were incorrect. We have revised the discussion.

(2) When discussing the DFT simulation that shows the increasing charge transfer to the interface region with the implication of greater adhesion, the paper "M. Wolloch, G. Levita, P. Restuccia, and M.C. Righi, Interfacial Charge Density and Its Connection to Adhesion and Frictional Forces, Physical Review Letters 121, 026804 (2018)" should be properly cited.

Response: Thank you for this insightful comment. The recommended article is very helpful for us to improve our manuscript. The recommended article is very helpful for us to improve our manuscript and was cited as the ref. [18] in the manuscript.

Wolloch et al. first discovered that both adhesion and friction are positively correlated with interfacial electronic charge redistribution, including interface with metallic, covalent, and physical bonds. This provides useful insight into understanding and regulating friction at the electronic level. Specifically, the more interface charges transferred, the stronger the interface adhesion, which is consistent with the results in our manuscript. Therefore, we have cited it in the sentence "The results shown in Fig. 4e indicate the charge (electron) transfer between graphene and the substrate increase with indentation (compress) distance This would lead to an increased adhesion of graphene on the substrate (Fig. 4e, right side)" on page 11 of the main text.

(3) More details on the calculations and justification of the model should be provided, namely: how was the common cell accommodating both the SiO₂ and graphene lattices constructed? How was strain distributed between the two? What was the reciprocal orientation? Why did the Authors choose an H termination of the SiO₂ surface? Is it representative of the experimental conditions?

Response: Thank you for the reminder. Providing more calculations and model details can help others understand and apply our work. We have included additional details below:

① The interface supercell consists of AB-stacked graphite (P63/MMC) and β -cristobalite SiO_2 (FD-3M). The lattice parameters of graphene are $a=b=2.46 \text{ \AA}$, $c=6.8 \text{ \AA}$, and the lattice parameters of SiO_2 are $a=b=c=7.16 \text{ \AA}$ ^{13, 14}. Graphene (001) surface and SiO_2 (001) surface are used for the calculations. The orientations of the graphene surface are $[-3 \ 1 \ 0]$, $[0 \ -3 \ 0]$, and lengths are 6.53 \AA and 7.40 \AA , respectively. The orientations of the SiO_2 surface are $[1 \ 0 \ 0]$, $[1 \ 0 \ 0]$, and lengths are 7.06 \AA and 7.06 \AA , respectively. The mismatch of the two orientations is 7.8% and 4.7%, respectively. We have added those details in the methods section.

② The tip and substrate for the experimental measurement are stored in atmosphere environmental conditions for a long time before the experiment, and the friction experiment was conducted in an air environment. The Si may react with H_2O and O_2 to oxidize and form SiO_2 ,¹⁵ and the SiO_2 surface has dangling bonds that easily react with H of water and form H passivated surfaces for SiO_2 .

Typos:

Page 3 line 7: *researche* → *research*

Response: Thank you for pointing out the errors. We are sorry for the negligence. The sentence "Although existing *researche* has provided some approaches..." has been revised to "Although existing studies have provided some approaches...".

We have conducted multiple reviews to eliminate the possible errors.

Page 12 "It can be deduced that the interaction between graphene and the substrate becomes stronger as the compression distance increases because the charge transferred at the interface from 0.18 e to 8.09 e" There seem to be some words missing in this sentence.

Response: Thank you for this insightful comment. We have reviewed this sentence and found that it may lack a description of the "compression distance". The complete sentence should be as follows: "It can be deduced that the interaction between graphene and the substrate becomes stronger as the compression distance of the interface graphene/substrate increases because the charge transferred at the interface from 0.18 e to 8.09 e". In addition, this sentence has been deleted and more detailed descriptions of charge transfer have been added to page 12 of the main text due to our revisions to the manuscript.

Reviewer #4 (Remarks to the Author):

Response: We would like to thank you for taking the time to review our manuscript. All reviewers' suggestion largely improves the presentation of our work.

REFERENCES

1. Lee JE, Ahn G, Shim J, Lee YS, Ryu S. Optical separation of mechanical strain from charge doping in graphene. *Nature Communications* **3**, 1024 (2012).
2. Zhang S, Hou Y, Li SZ, Liu LQ, Zhang Z, Feng XQ, Li QY. Tuning friction to a superlubric state via in-plane straining. *P Natl Acad Sci USA* **116**, 24452-24456 (2019).
3. Hong Y, Wang SM, Li Q, Song X, Wang ZG, Zhang X, Besenbacher F, Dong MD. Interfacial icelike water local doping of graphene. *Nanoscale* **11**, 19334-19340 (2019).
4. Kumar S, Shakya J, Mahanta T, Kanjilal D, Mohanty T. Substrate-assisted Fermi level shifting of CVD graphene by swift heavy ions. *Surf Interfaces* **28**, 101625 (2022).
5. Melitz W, Shen J, Kummel AC, Lee S. Kelvin probe force microscopy and its application. *Surf Sci Rep* **66**, 1-27 (2011).
6. Wang R, Wang SN, Zhang DD, Li ZJ, Fang Y, Qiu XH. Control of Carrier Type and Density in Exfoliated Graphene by Interface Engineering. *Acs Nano* **5**, 408-412 (2011).
7. Cahangirov S, Ciraci S, Özçelik VO. Superlubricity through graphene multilayers between Ni(111) surfaces. *Physical Review B* **87**, 205428 (2013).
8. Socoliuc A, Bennewitz R, Gnecco E, Meyer E. Transition from stick-slip to continuous sliding in atomic friction: Entering a new regime of ultralow friction. *Phys Rev Lett* **92**, 134301 (2004).
9. Lang HJ, Zou K, Chen RL, Huang Y, Peng YT. Role of Interfacial Water in the Tribological Behavior of Graphene in an Electric Field. *Nano Letters* **22**, 6055-6061 (2022).
10. Derjaguin BV, Muller VM, Toporov YP. Effect of Contact Deformations on the Adhesion of Particles. *Prog Surf Sci* **45**, 131-143 (1994).
11. Wagner K, Cheng P, Vezenov D. Noncontact Method for Calibration of Lateral Forces in Scanning Force Microscopy. *Langmuir* **27**, 4635-4644 (2011).
12. Ogletree DF, Carpick RW, Salmeron M. Calibration of frictional forces in atomic force microscopy. *Rev Sci Instrum* **67**, 3298-3306 (1996).
13. Chung DDL. Review graphite. *J Mater Sci* **37**, 1475-1489 (2002).
14. Demuth T, Jeanvoine Y, Hafner J, Angyán JG. Polymorphism in silica studied in the local density and generalized-gradient approximations. *J Phys-Condens Mat* **11**, 3833-3874 (1999).
15. Chen L, Xiao C, He X, Yu BJ, Kim SH, Qian LM. Friction and Tribochemical Wear Behaviors

- of Native Oxide Layer on Silicon at Nanoscale. *Tribology Letters* **65**, (2017).
16. Gupta RP. Electronic-Structure of Crystalline and Amorphous-Silicon Dioxide. *Physical Review B* **32**, 8278-8292 (1985).
 17. Revesz AG, Hughes HL. The Structure of Thermally Grown Noncrystalline SiO₂-Films on Silicon. *J Non-Cryst Solids* **71**, 87-94 (1985).
 18. VigneMaeder F, Sautet P. Theoretical study of hydroxylated and dehydroxylated surfaces of a cristobalite model of silica. *J Phys Chem B* **101**, 8197-8203 (1997).
 19. Iarlori S, Ceresoli D, Bernasconi M, Donadio D, Parrinello M. Dehydroxylation and silanization of the surfaces of β -cristobalite silica:: An ab initio simulation. *J Phys Chem B* **105**, 8007-8013 (2001).
 20. Rozanska X, Delbecq F, Sautet P. Reconstruction and stability of β -cristobalite 001, 101, and 111 surfaces during dehydroxylation. *Phys Chem Chem Phys* **12**, 14930-14940 (2010).
 21. Carpick RW, Ogletree DF, Salmeron M. A general equation for fitting contact area and friction vs load measurements. *J Colloid Interf Sci* **211**, 395-400 (1999).
 22. Qi YZ, Liu J, Zhang J, Dong YL, Li QY. Wear Resistance Limited by Step Edge Failure: The Rise and Fall of Graphene as an Atomically Thin Lubricating Material. *Acs Appl Mater Inter* **9**, 1099-1106 (2017).
 23. Popov VL. *Contact mechanics and friction*. Springer (2010).
 24. Wolloch M, Levita G, Restuccia P, Righi MC. Interfacial Charge Density and Its Connection to Adhesion and Frictional Forces. *Phys Rev Lett* **121**, 026804 (2018).
 25. Niesner D, Fauster T. Image-potential states and work function of graphene. *J Phys-Condens Mat* **26**, 393001 (2014).

REVIEWER COMMENTS

Reviewer #1 (Remarks to the Author):

In the revision, Peng and their coworkers satisfactorily responded to my comments. I have reviewed the revised version and found that all answers/responses have been addressed carefully. Moreover, the authors improved the discussion with clear justification. The details of SKPM, Raman spectroscopy, and simulations are technically correct and coherent with the manuscript. In my opinion, the manuscript can be accepted in the present format.

Reviewer #2 (Remarks to the Author):

The authors have addressed all my concerns in the revisions. Hence, I am happy to recommend the publication of this manuscript.

Reviewer #3 (Remarks to the Author):

The Authors replies to our comments are satisfactory, therefore we recommend this paper for publication.

Reviewer #4 (Remarks to the Author):

Reply to Reviewers:

We would like to express our sincere gratitude for all the reviewers' valuable comments and insightful feedback throughout the review process. Your constructive suggestions greatly contributed to enhancing the quality and clarity of our work. We are truly grateful for the time and expertise you dedicated to reviewing our manuscript, and we are pleased to hear that it has been accepted.

Thank you again for your support and for the opportunity to share our research.

Yours sincerely,

Haoyang Su, Honglin Zhang, Junhui Sun, Haojie Lang, Kun Zou, Yitian Peng

Su and coworker presented the frictional characteristics of graphene layer/s in ambient conditions. The experimental study is supported by DFT simulation for charge migration and contact mechanics. The interesting concept authors bring the pre-rubbing surface to tune down the friction force and its revival under optimised pressurised rubbing. The presentation of results is clean without ambiguity, but a weak discussion is presented. Therefore, I am considering the article for acceptance after major revisions. Here are a few technical considerations for the authors.

1. Typo at page three.
2. On page 5, "crater" " graphene under deformed structure is an interesting study. Can the author figure out the possibility of strain (experimentally) through Raman spectroscopic mapping or high-resolution imaging in the graphene over deformed silica? I will recommend following article to get useful information
<https://doi.org/10.1002/sml.202104487>
3. The SKPM technique reveals the difference in surface charge potential between pre- and post-rubbed areas. This is the critical part of the manuscript, and the author just mentioned its superficial information. The author must quantify the outcomes of SKPM following its calibration.
4. Three factors could be responsible for lower friction force: interfacial adhesion (value is missing), strain (possibly tension missing value) and charge accumulation (not mentioned). The author must present some analytical relation with these quantitative values.
5. What kind of bond formation/hybridisations/proximity effect occurred during rubbing?
6. Page 12: It is unclear why friction force increases if graphene is strongly adhered after pre-rubbing at lower substrate pressure.
7. In figure 4 (d), what is the charge distribution associated to blue color. Is it rich in electrons or holes concentration?
8. I am not able to find the normal and torsional stiffness of the cantilever. Please add the calibration values and its procedure.